# OpenPRM: Building Open-domain Process-based Reward Models with Preference Trees

**Kaiyan Zhang**[1]    **Jiayuan Zhang**[2]    **Haoxin Li**[1]    **Xuekai Zhu**[3]    **Ermo Hua**[1]
**Xingtai Lv**[1]    **Ning Ding**[1]    **Biqing Qi**[4]    **Bowen Zhou**[1,4] [*]

[1] Tsinghua University    [2] Beihang University    [3] Shanghai Jiao Tong University
[4] Shanghai Artificial Intelligence Laboratory

## ABSTRACT

Scaling inference-time computation is increasingly seen as the next frontier in scaling laws for large language models. Previous work in mathematics and coding has demonstrated the remarkable potential for inference-time scaling. During such scaling, fine-grained supervision through process-based reward models (PRMs) is essential for enhancement. However, exploration of inference-time scaling and PRMs in open-domain problems remains limited, where lacking exact answers and obtaining process supervision prove challenging. In this paper, we explore the construction of PRMs for open-domain tasks, specifically for instruction-following tasks. Utilizing existing outcome-based reward models (ORMs), we develop sentence-level preference trees based on the prefix similarity of parallel sampled candidates from datasets like UltraFeedback. This setup allows us to derive weak supervision for processes via back-propagation from outcome-level rewards. Subsequently, we integrate ORMs and PRMs under the same pairwise ranking objectives, resulting in our newly developed reward models, named Open-PRM. This approach significantly enhances the scalability of process-level supervision in open domains at minimal cost. We assess the performance of OpenPRM across various reward benchmarks, demonstrating its competitive edge over traditional ORMs in open domains and PRMs in specialized domains. Additionally, we investigate the scalability of inference-time computation for open-domain instructions. Our results highlight the limitations of ORMs' scalability, while OpenPRM shows superior performance in scaled settings. Despite these advances, achieving automatic fine-grained supervision for open-domain inference-time scaling remains a substantial challenge. We hope these findings will spur further development of process supervision reward models in open-domain scenarios.

## 1 INTRODUCTION

Large language models (LLMs) such as GPT-4 (Achiam et al., 2023), Llama (Touvron et al., 2023; Dubey et al., 2024), and Gemini (Team et al., 2023; Reid et al., 2024) have garnered interest across various fields due to their robust performance in numerous tasks and domains. The development of LLMs involves an official process that includes pre-training on a large-scale unlabeled corpus (Brown, 2020) followed by post-training using labeled instructions derived from real-world applications. The post-training phase is further categorized into supervised fine-tuning and reinforcement learning from human or model feedback (Ouyang et al., 2022), a process known as alignment (Ji et al., 2023; Shen et al., 2023). During this phase, reward models are crucial as they act as human proxies, providing feedback on model behavior and adjusting the models to better align with human values (Bai et al., 2022). Although current popular alignment algorithms like Direct Preference Optimization (DPO) (Rafailov et al., 2024) implicitly incorporate the rewarding process within the loss function, reward models still play a significant role in ensuring long-term alignment through methods such as online and iterative DPO (Xiong et al., 2023; Guo et al., 2024; Pang et al., 2024) and rejected sampling (Dong et al., 2023; Liu et al., 2023; Khaki et al., 2024).

---

[*]Corresponding Author (zhoubowen@tsinghua.edu.cn)

In addition to the training process, reward models are crucial for enhancing the performance of LLMs during inference-time (Khanov et al., 2024; Deng & Raffel, 2023). Unlike the scaling laws applied to compute during pre-training (Kaplan et al., 2020; Hoffmann et al., 2022), there is a trend towards scaling inference-time compute through extensive searches in the decoding space (Snell et al., 2024; Brown et al., 2024). Reward models play a significant role in pruning the search space and ultimately selecting the most accurate answers (Welleck et al., 2024).

Recent studies indicate that outcome-level reward models fail to apply scaling laws during repeated sampling (Brown et al., 2024), particularly due to their coarse granularity on challenging tasks. Consequently, many researchers are exploring the use of more fine-grained reward models, such as process-level (Uesato et al., 2022; Lightman et al., 2023) or token-level (Deng & Raffel, 2023), to enhance search performance in specialized domains like mathematics, coding, and reasoning tasks (Wang et al., 2024b; Havrilla et al., 2024; Xin et al., 2024; Yuan et al., 2024; Chen et al., 2024a; Setlur et al., 2024; Xie et al., 2024). These efforts often follow the paradigm established by AlphaGo (Silver et al., 2016), integrating LLMs with Monte Carlo Tree Search (Coulom, 2006; Kocsis & Szepesvári, 2006) and a value function, analogous to a reward model. However, process-level reward models are typically tailored for specific tasks and exhibit limited generalizability in open-domain applications such as writing and chat. Moreover, there is scant research on developing process-level reward models for open-domain contexts, primarily due to the high cost of annotation.

Currently, outcome-level reward models (ORMs) are evolving rapidly (Wang et al., 2024a; Cai et al., 2024; Wang et al., 2024e;c; Vu et al., 2024), prompted by the emergence of datasets and benchmarks such as Ultrafeedback (Cui et al., 2023) and RewardBench (Lambert et al., 2024), where open-source reward models trend to outperform proprietary ones. ***This development raises the question of whether process-level reward models (PRMs) can be constructed from instance-level rewards using a weak-to-strong framework*** (Burns et al., 2023).

In this paper, we propose the development of PRMs in open-domain, leveraging existing ORMs to provide fine-grained supervision. Our contributions are summarized as follows: (1) We analyze the potential for extending open-domain ORMs to PRMs, elucidating the characteristic and relationship between them. This analysis inspires our proposal to develop PRMs with outcome-level supervision through building preference trees with key process. (2) We integrate the modeling of PRMs and ORMs under a unified objective and develop OpenPRM. By leveraging only existing ORMs and employing repeated sampling on prompts, we enhance the performance of ORMs, achieving a 3~5% improvement on RewardBench. (3) We further evaluate OpenPRM across various downstream applications, including different inference-time scaling settings. Our findings show that ORMs struggle to provide effective supervision, while our proposed OpenPRM outperforms previous RMs under these scaling conditions. We also observe that there is a significant journey ahead to fully realize the potential of RMs in open-domain tasks for test-time scaling.

## 2 PRELIMINARY

### 2.1 REWARD MODELING

Reward models play a crucial role in large language models by aligning model outputs with desired human preferences (Wang et al., 2023). There are primarily two types of reward models based on the granularity of the supervision signal: outcome-level and process-level reward models. We will introduce the development of these two methods as follows.

**Outcome-level Reward Model (ORM)** ORMs are commonly used for preference learning, particularly after supervised fine-tuning in InstructGPT, where they serve as a proxy for human feedback to model generations (Ouyang et al., 2022; Lee et al., 2023). Although many studies explore reward model-free preference learning, such as direct preference optimization (DPO) (Rafailov et al., 2024), which implicitly models the reward within the policy model training, ORMs continue to be instrumental in further model improvements. This includes applications in online or iterative DPO (Guo et al., 2024; Pang et al., 2024) and rejection sampling (Liu et al., 2023; Dong et al., 2024).

The primary methods for obtaining ORMs involve preparing pairwise responses with preferences (e.g., chosen and rejected) and fine-tuning instructed models using ranking loss (Ouyang et al., 2022; Dong et al., 2024). Some studies also consider DPO models as reward models (Lambert et al., 2024),

Table 1: A comparison with the most related works on process-level reward models.

| Name | Data Acquisition | | | | | Training & Inference | | | Release | |
|------|--------|------|----------|------------|------|----------|-----------|--------|------|-------|
|      | **Domain** | **Task** | **Backbone** | **Annotation** | **Size** | **Labeling** | **Objective** | **Search** | **Data** | **Model** |
| DEEPMIND PRM (Uesato et al., 2022) | Math | GSM8K | N/A | Human | 10k | 0/1 | N/A | BoN | ✗ | ✗ |
| OPENAI PRM (Lightman et al., 2023) | Math | MATH | GPT-4 | Human | 800k | -1/0/1 | CE Loss | BoN | ✓ | ✗ |
| TS-LLM (Feng et al., 2023) | Math Decision | GSM8K,GAME24, ProntoQA,RLHF, Chess Endgame | LLaMA2-7B | Golden Ans. | ~150k | 0~1 | MSE Loss | MCTS-$\alpha$ +Rollout 100×17 leaf | ✗ | ✓ |
| MATH-SHEPHERD (Wang et al., 2024b) | Math | GSM8K | LLemma-7B | Golden Ans. | 445k | 0/1 | CE Loss | MCTS 16×64 leaf | ✓ | ✓ |
| GLORE (Havrilla et al., 2024) | Math | GSM8K | Llama2-7B | Golden Ans. | N/A | 0/1 | CE Loss | BoN | ✗ | ✗ |
| MIPS (Wang et al., 2024f) | Math/Code | GSM8K, MATH MBPP | PaLM 2-S/L | Golden Ans. | ~14k | 0~1 | MSE Loss | MCTS 32×32 leaf | ✗ | ✗ |
| MCTS-DPO (Xie et al., 2024) | Math Common -sense | GSM8K,MATH ARC,AI2Science, OpenBookQA, CommonSenseQA | Mistral-7B | Golden Ans. | ~24k | 0~1 | MSE Loss | MCTS 4/5 leaf | ✗ | ✗ |
| SUPER MARIO (Chen et al., 2024a) | Math | GSM8K,MATH | DeepSeek-MathBase-7B | Golden Ans. | 15k | 0~1 | MSE Loss | MCTS 5 leaf | ✓ | ✗ |
| STEP-DPO (Lai et al., 2024) | Math | MetaMath, MMIQC | Qwen2-7B/72B | LLM | 10k | 0/1 | Ranking Loss | BoN | ✓ | ✗ |
| OMEGAPRM (Luo et al., 2024) | Math | MATH | Gemini Pro | Golden Ans. | 1.5m | 0~1 | MSE Loss | MCTS | ✗ | ✗ |
| REST-MCTS* (Zhang et al., 2024) | Math | MATH | Mistral-7B | Golden Ans. | ~700k | 0~1 | MSE Loss | MCTS 3 leaf | ✓ | ✗ |

though the effectiveness of these models in iterative optimization still requires further exploration. Additionally, self-play learning (Chen et al., 2024b; Tao et al., 2024) has recently been applied to continually improve reward models, presenting a promising method to enhance the capabilities of ORMs autonomously (Wang et al., 2024c). Our approach on OpenPRM can also be considered a method for continuously improving ORMs through their own annotations.

**Process-level Reward Model (PRM)** The primary challenge with ORMs is their coarse-grained nature of rewards; even if the final answer is correct, errors may still exist within the solution steps. To address this, there is a growing trend to develop more fine-grained, process-level RMs. The main challenge in developing PRMs lies in obtaining accurate supervision signals for each process within a solution. There are three main approaches: 1) Human Annotation: This method requires experts to annotate each process step as neutral, bad, or good. While human annotation can provide precise process supervision, it is difficult to scale and very costly (Uesato et al., 2022; Lightman et al., 2023). 2) Golden Answer.: For mathematical or coding problems, accurate final answers or feedback from exact matching or interpreters are available. Common methods involve computing the probability of nodes along the path toward the final, accurate answers, integrating with Monte Carlo Search methods (Wang et al., 2024b; Havrilla et al., 2024; Luo et al., 2024). 3) Model-based Judgment or Reward Models: The final approach involves obtaining rewards from model-based judgments or reward models (Lai et al., 2024). Some research utilizes outcome-level rewards to estimate process rewards (Lu et al., 2024), reducing the high costs associated with extensive sampling.

For training PRMs, two main methods are used, depending on the data format required: 1) Single Sample: Each step in the solution process is labeled, and losses such as Cross-Entropy (CE) (Lightman et al., 2023; Wang et al., 2024b; Havrilla et al., 2024) and Mean Squared Error (MSE) (Feng et al., 2023; Wang et al., 2024f) loss are typically used. 2) Pair Sample: Each question is associated with chosen and rejected processes, and pairwise ranking loss is employed (Lai et al., 2024). This method is typically used in the training ORMs (Dong et al., 2024). We provide a detailed survey in Table 1. Currently, many ORMs in open domains benefit from the development of benchmarks (Lambert et al., 2024; Wang et al., 2024e). However, the true effectiveness of these ORMs and the feasibility of developing PRMs from ORMs are still subjects of ongoing exploration.

## 2.2 DERIVATION OF PRM FROM ORM

ORM is typically trained to predict the quality of the final outcome, while PRM supervises the intermediate steps of the process. The modeling of ORM can be formalized with a pairwise ranking

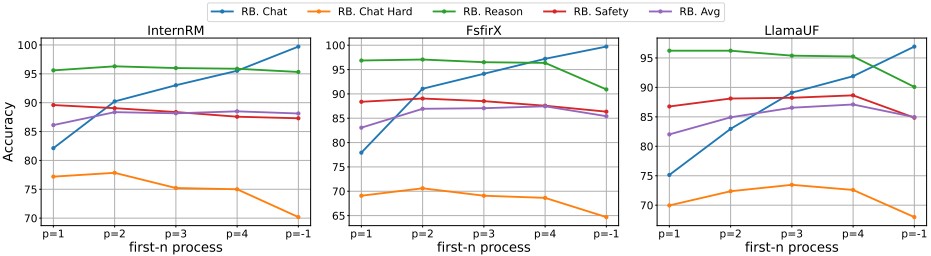

Figure 1: Results of reward models based on rewarding processes of varying lengths within the evaluated content. The results indicate that the initial segments of responses are particularly critical for challenging tasks such as Chat-Hard and Reasoning, where longer content holds length bias.

loss function based on the outcome-level feedback (Ouyang et al., 2022):

$$\mathcal{L}_{\text{ORM}}(\theta) = -\mathbb{E}_{(x,y_c,y_r)\sim D}\left[\log\left(\sigma\left(r_\theta(x, y_c) - r_\theta(x, y_r)\right)\right)\right] \tag{1}$$

Here, $r_\theta(x, y_c)$ and $r_\theta(x, y_r)$ are the model's scores for the chosen and rejected outcomes, respectively. While ORM performs well in outcome-based supervision, it has limitations when applied to process-level supervision, due to the cumulative error effect. This effect arises because ORM focuses on the final result, neglecting errors in intermediate steps that can propagate through the sequence and affect the final outcome (Lightman et al., 2023).

Previous works (Uesato et al., 2022; Havrilla et al., 2024) have shown that in cases where the base model for response generation is sufficiently strong and the task is relatively simple, the cumulative error may be negligible. In such scenarios, ORM can be effectively used as a substitute for PRM.

However, for more complex tasks, the cumulative error can be significant, necessitating additional process-level supervision. To address this issue, we propose a joint modeling approach that integrates both ORM and PRM by introducing supervision at key process of answer. Specifically, we identify the most divergent process between chosen and rejected outcomes, denoted as $p_c$ and $p_r$, and introduce additional supervision at these critical points. The loss function can be defined as:

$$\mathcal{L}(\theta) = \mathcal{L}_{\text{ORM}}(\theta) + \lambda\mathcal{L}_{\text{PRM}}(\theta) \tag{2}$$

where $\lambda$ is a hyper-parameter that balances the outcome-based and process-based losses. The process-based loss $\mathcal{L}_{\text{PRM}}$ supervises the divergent steps $p_c$ and $p_r$, and can be expressed as:

$$\mathcal{L}_{\text{PRM}}(\theta) = -\log\left(\sigma\left(r_\theta(x, p_c) - r_\theta(x, p_r)\right)\right) \tag{3}$$

Here, $r_\theta(x, p_c)$ and $r_\theta(x, p_r)$ represent the scores for the critical steps in the chosen and rejected sequences, respectively. By focusing on these key steps, the cumulative error is mitigated as early errors are corrected at critical junctures. We provide more details in Appendix A. In conclusion, the above analysis leads to the following theorem:

**Theorem 1** *Given a dataset $D$ consisting of pairs of responses $(y_c, y_r)$ with outcome-based preferences $y_c > y_r$, and a learned outcome-based reward model $r_\theta(x, y)$, the cumulative error of process supervision can be significantly reduced. This is achieved by identifying the key divergent steps $(p_c, p_r)$, such that $\Delta(p_c, p_r)$ is maximized, and incorporating these steps into a joint modeling framework. Thus, under this framework, the cumulative error in process supervision decreases as the discrepancy $\Delta(p_c, p_r)$, supervision strength $S$, and model sensitivity $\gamma$ increase.*

## 2.3 EMPIRICAL EVALUATION OF OPEN-DOMAIN ORM IN PROCESS ASSESSMENT

In this section, we conduct an empirical evaluation of open-domain ORMs as process evaluators and examine some unique phenomena associated with their performance. Specifically, we assess popular reward models such as FsfirX (Wang et al., 2024a), InternRM (Cai et al., 2024), and UltraFeedback (Cui et al., 2023) (trained on the Llama-3 8B model (Dubey et al., 2024)) using RewardBench (named RB) (Lambert et al., 2024). We focus on the primary categories of RB, including Chat, Chat-Hard, Reasoning, and Safety tasks. We provide more details about experiments in § 4.1.

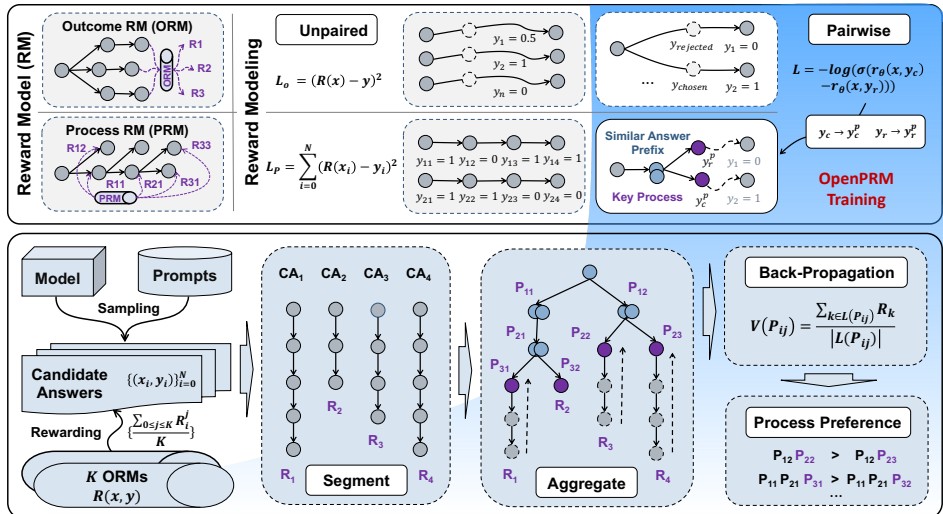

Figure 2: This figure illustrates the differences between outcome-level and process-level reward models in the top left, including their common training strategies with paired and unpaired data. The training of OpenPRM is depicted on the right and below (mainly blue area).

As illustrated in Figure 1, a consistent trend is observed across various reward models, indicating that RMs perform better on simpler Chat tasks as the evaluated process lengthens. However, this effect reverses in more challenging Chat and Reasoning tasks, where accuracy first increases and then decreases as the length of the evaluated process extends. In conclusion, these results support our Theorem 1, which posits that ORMs can function as PRMs, but their performance deteriorates due to cumulative errors, particularly in harder tasks.

## 3  METHODOLOGY

### 3.1  PROCESS-LEVEL PREFERENCE TREE

To obtain process-level rewards, we construct process-level preference trees using readily available outcome reward models. The pipeline for building preference trees consists of three steps:

**Step 1.  Repeated Sampling on Prompts** We initially prompt open-source language models to generate a large number of parallel candidate responses through repeated sampling. To ensure broad representation, we primarily include models at the 7B and 70B parameter levels.

**Step 2.  Aggregation on Sentences** For each output, we segment it into a collection of sentences and construct a tree using depth-first search algorithms. We calculate the edit distance (Ristad & Yianilos, 1998) between sentences from different outputs and merge sentences into a single node based on a predefined threshold. This helps us reduce the cost of building prefix trees.

**Step 3. Backpropagation on Rewards** Once outputs with their respective rewards are segmented into sentence collections, which serve as nodes in the preference tree, we designate the outcome rewards for the leaf nodes. For each process node, we compute the process-level rewards using backpropagation. Given the rewards of the leaf nodes, $R_k$ (outcome-level rewards), we can compute the rewards of the inner nodes, $P_{ij}$, using backpropagation, as detailed in Monte Carlo Tree Search. Notably, the rewards of the inner nodes, denoted as $V(P_{ij})$, can be calculated using the formula

$$V(P_{ij}) = \frac{\sum_{k \in L(P_{ij})} R_k}{|L(P_{ij})|},$$

where $L(P_{ij})$ represents the set of all leaf nodes descending from $P_{ij}$.

**About Rationality of Process Aggregation** Unlike previous works in mathematics and coding that reuse partial answers for subsequent answer generations (Lightman et al., 2023; Luo et al., 2024),

our method involves directly sampling a large number of candidates and merging identical sentences, akin to state aggregation in Monte-carlo algorithms (Hostetler et al., 2014; Jang et al., 2021). This approach enables the RMs to learn high-level actions and logic within the shared sentences. We provide a real example of question-answering for reference in the Appendix B.3.

## 3.2 PROCESS-LEVEL REWARD MODELING

During the development of OpenPRMs, we enhance the models by integrating rewards and domains, aiming to create more robust process-level reward models from the following two perspectives:

**Mixture of Rewards** Considering the completeness of outputs, output-level rewards serve as specific instances of process-level rewards, where the output encompasses the entire process. Therefore, we blend rewards from both the process and output levels to develop more robust reward models.

**Mixture of Domains** Existing process-level reward models predominantly focus on domains such as mathematics, and reasoning tasks, which provide certain answers for supervision. To leverage the strengths of these domains, we also integrate them with general domain preferences to enhance the versatility and applicability of OpenPRMs. We provide details about dataset in Appendix B.1.

At the training stage, we treat all preference data as a pairwise ranking task. This involves using the input prompt along with chosen and rejected completions (including both process and output). Using this unified format, we train the PRM with the Bradley-Terry objective, as defined in Equations 1 and 3. This formulation ensures consistent training across both process- and outcome-level datasets.

## 3.3 APPLICATION OF PROCESS-LEVEL REWARD MODELS

**Best-of-N Sampling** At inference time, we can generate a large number of candidate answers for given questions. Subsequently, we can determine the final answer through a majority vote (James, 1998) (referred to as *Vote@N*); however, this method is primarily applicable to questions that require exact answers, such as those found in mathematics and reasoning tasks. For open-ended questions, it is more common to apply reward models to all answers and select the one with the highest rewards, a method known as best-of-N sampling (Stiennon et al., 2020) (*BoN@N*). When implementing process-level reward models in the BoN context, there are two approaches to computing rewards: one approach treats the outcome as a special process and computes rewards directly on the outcome, while the other calculates rewards for each process and selects the minimal one (Lightman et al., 2023; Wang et al., 2024b) to derive the outcome rewards.

**Process-level Decoding** Another significant application of PRMs is in the decoding phase. By evaluating the generated process, we can expand the beam search strategy (Sutskever et al., 2014) from token-level to process-level. As a result, we maintain N sentences at each step and reward each sentence during generation until the completion of the answer, a technique termed process-level beam search (*PBS@N*). Additionally, we can integrate advanced operations akin to those employed in Monte Carlo Tree Search (MCTS) (Browne et al., 2012), such as simulation, retrospection, and memory functions. However, these operations may extend the required processing time and lead to increased inference costs. Previous research (Chen et al., 2024a; Snell et al., 2024) has indicated that *PBS@N* can achieve performance comparable to MCTS but at a reduced cost.

## 4 EXPERIMENTAL SETUP

### 4.1 DATASET

In developing OpenPRM, we first construct extensive preference trees based on open-domain instruction dataset, as described in § 3.1. This construction utilizes the UltraFeedback (Cui et al., 2023) and ScienceQA (Lu et al., 2022) datasets, which provide a highly diverse and high-quality range of instructions. Additionally, we incorporate the MATH (Hendrycks et al., 2021) dataset to further enhance the math reasoning capabilities of our reward system. For each prompt within this instruction pool, we sample 64 candidate responses from Llama-3 models (Dubey et al., 2024).

## 4.2 MODELS

**Reward Models.** We compare our reward models with state-of-the-art (SOTA) open-source reward models. Due to concerns with inference efficiency, we primarily evaluate classifier-based models, which perform comparably to generative models but are more scalable. We compare our models with ORMs, such as FsfairX (Dong et al., 2024), Eurus (Yuan et al., 2024), and UltraRM (Cui et al., 2023), and PRMs like TS-LLM (Chen et al., 2024a), MathShepherd (Wang et al., 2024b).

**Chat Models.** We assess the effectiveness of our reward models using state-of-the-art open-sourced chat models, including Llama-3.1-8B-Instruct and Llama-3.1-70B-Instruct (Dubey et al., 2024), and Mistral-Nemo-Instruct-2407 [1]. The latter can be regarded as an out-of-distribution evaluation.

## 4.3 EVALUATIONS

**Evaluation of Reward Models.** Given the lack of established benchmarks for evaluating process-based reward models, we primarily compare our process-based models against established outcome-based reward benchmarks, such as UltraFeedback (Cui et al., 2023) and RewardBench (Lambert et al., 2024). RewardBench is designed to assess the capabilities and safety of reward models across four categories: Chat, Hard Chat, Reasoning, and Safety. We employ the primary dataset from RewardBench to evaluate the out-of-domain generalization capabilities of our reward models. We evaluate the effectiveness of process supervision of reward models solely on the test set of PRM800k (Lightman et al., 2023), which features high-quality human annotations. Especially, we evaluate PRMs using specific aggregation strategies, such as selecting the minimal reward across steps. Detailed descriptions of the different aggregation strategies are provided in the Appendix D.1.

**Evaluation of Chat Models.** To comprehensively evaluate the impact of reward models on chat models, we test the chat models across a variety of benchmarks, primarily referencing the Open LLM Leaderboard [2]. This evaluation includes benchmarks in: 1) Instruction following tasks such as Alpaca Eval 2 (Dubois et al., 2024) and IFEval (Zhou et al., 2023); 2) General domain tasks such as MixEval Hard (Ni et al., 2024), MMLU-Pro (Wang et al., 2024d), and GPQA (Rein et al., 2023); 3) Specific math domain tasks like MATH500 (Lightman et al., 2023). Additional details about these evaluation tasks and methodology are provided in Appendix C.

## 4.4 SETTINGS FOR INFERENCE

During inference, we primarily evaluate two methods: majority vote and best-of-N. For the best-of-N method, we adhere to the following protocol (Chen et al., 2021): initially, we sample N responses, where N is set to 128. We then sample K responses from these, repeating the process M times to average the results. The values for K range from 1, 2, 4, 8, 16, 32, 64, to 128, and M is set to 5. This approach allows us to reduce inference costs and achieve robust results through multiple averaging. For process-based beam search, we set the beam size to $\sqrt{N}$ to maintain an approximately equivalent decoding cost with best-of-N, as described in (Snell et al., 2024).

## 5 EXPERIMENTAL RESULTS

### 5.1 RESULTS OF REWARD BENCHMARKS

As shown in Table 2, we compare OpenPRMs with standard ORMs and specialized PRMs across both general outcome-based and specific process-reward benchmarks. Based on the results, we can draw the following conclusions:

**OpenPRMs outperform ORMs** Utilizing off-the-shelf ORMs and corresponding preference datasets, we have developed advanced reward models that demonstrate superior performance on RewardBench, particularly in the Chat Hard and Reasoning tasks. Additionally, the process-based preferences built upon our method consistently enhance the performance of the base reward models,

---

[1] https://mistral.ai/news/mistral-nemo/
[2] https://hf.co/spaces/open-llm-leaderboard/open_llm_leaderboard

Table 2: Results of outcome-level and process-level reward models on instance-level reward benchmarks. Models marked with an asterisk (*) were trained using data compiled by our team.

| Model / Task | Training Data | UltraFeedback Test | RewardBench | | | | | PRM800k Test |
| --- | --- | --- | --- | --- | --- | --- | --- | --- |
| | | | Overall | Chat | Chat Hard | Safety | Reasoning | |
| *Open-Source ORMs* | | | | | | | | |
| ULTRARM-13B | UltraFeedback (UF) | 74.8 | 68.5 | 96.4 | 55.5 | 59.9 | 62.4 | 50.8 |
| LLAMA-3-8B* | UF Binarized | **77.8** | 84.8 | 97.5 | 66.9 | 85.5 | 89.2 | 51.8 |
| EURUS-RM-7B | UltraInteract | 73.5 | 82.8 | 98.0 | 65.6 | 81.4 | 86.3 | 60.6 |
| FSFAIRX-7B | Mixture Preference | 74.5 | 84.4 | **99.4** | 65.1 | 86.8 | 86.4 | 53.3 |
| INTERN2-7B-RM | Unknown | 77.4 | **87.6** | 99.2 | 69.5 | **87.2** | **94.5** | **61.0** |
| LLAMA-3-8B* | HelpSteer (HS) 2 | 71.8 | 86.8 | 95.3 | **76.8** | 85.9 | 89.2 | 53.7 |
| *Open-Source PRMs (Merely Math Domain)* | | | | | | | | |
| MS-7B-PRM | Math-Shepherd | 53.5 | 56.6 | 62.3 | 51.3 | 39.6 | 73.2 | 56.9 |
| TS-LLM | Unknown | 52.7 | 57.6 | 66.8 | 50.0 | 55.3 | 58.4 | 57.7 |
| LLAMA-3-8B* | Math-Step-DPO | **70.2** | **73.2** | **98.0** | **58.8** | **59.9** | **76.1** | **61.3** |
| OPENPRM (FsfairX) | UF Tree + HS 2 | 72.8 | 89.4 | 95.5 | 81.1 | 88.7 | 92.1 | 64.3 |
| OPENPRM (InternRM) | UF Tree + HS 2 | **78.5** | **91.1** | **98.0** | **81.6** | **89.5** | **95.1** | **68.1** |

showcasing their generalization capabilities. These findings validate the effectiveness of our preference tree construction strategy discussed in § 3.1. Moreover, the results substantiate our ability to enhance weaker existing models, achieving weak-to-strong generalization (Burns et al., 2023).

**OpenPRMs outperform specific PRMs** Beyond outcome-level reward benchmarks, we also compare OpenPRMs with publicly available PRMs, which predominantly originate from the math domain, as many previous PRMs are not available. We present some results evaluated using Math-Shepherd (Wang et al., 2024b), TS-LLM (Feng et al., 2023) and Llama-3 trained on (Lai et al., 2024). Due to the domain gap, these math-specific PRMs underperform in open-domain benchmarks, whereas OpenPRMs demonstrate superior performance even on tasks like PRM800k.

## 5.2 RESULTS OF APPLICATIONS IN DECODING

To validate the effectiveness of PRMs, we evaluated OpenPRM under various decoding settings across multiple popular open-domain tasks, comparing strategies such as majority vote, best-of-N, and process-level beam search. We summarize the experimental results of OpenPRM as follows:

**OpenPRM Performs Effectively with BoN and PBS** As illustrated in Table 3, OpenPRM achieves superior performance in both BoN@16 and PBS@4 compared to Vote@16 with the Llama-3.1-8B and 70B models across nearly all tasks. These results confirm the effectiveness of OpenPRM. Additionally, even the out-of-distribution models, such as Mistral-Nemo (compared to the Llama series), validate the advantages of OpenPRM. We also observed that beam search algorithms outperform BoN, benefiting from the fine-grained evaluation of processes. However, further exploration of decoding strategies (like MCTS) in open-domain settings will be necessary in the future.

**Scaling Inference-Time Achieves a High Upper Boundary** We further analyze the results of scaling inference-time by progressively increasing sampling times from 1 ($2^0$) to 128 ($2^7$). The results depicted in Figures 3 demonstrate that the models can achieve exceptional performance with optimal reward models (refer to coverage@N and pass@N settings (Chen et al., 2021)). The coverage accuracies nearly reach 100% on most open-domain tasks. These findings in open-domain tasks are consistent with prior studies in mathematical and coding domains (Brown et al., 2024; Bansal et al., 2024), suggesting the emergence of a new scaling law at inference time in open-domain as well. We also include the sampling curves for the Llama 70B and Mistral model in Appendix D.4.

**OpenPRMs Optimize Inference-Time Utilization** Compared to the coverage accuracy depicted by the red curve, nearly all the reward models struggle to scale as inference-time increases. This indicates that significant advancements are still required to develop more effective reward models for inference-scaling. Among these previous models, InternRM performs best on most tasks on average, notably on MMLU-Pro and IFEval, though it still lags significantly behind the coverage

Table 3: Results of majority vote, best-of-N sampling, and process-level search on open-domain tasks. The findings from BoN@16 and PBS@16 demonstrate the effectiveness of OpenPRMs. Notably, we have reproduced part of the results, taking into account differences in dataset usage.

| Model / Task | | Alpaca Eval 2 | | MixEval | IFEval | GPQA | MMLU-P* | MATH* |
|---|---|---|---|---|---|---|---|---|
| | | LC% | WC% | Acc | Avg. | Acc | Acc | Acc |
| GPT-4o (0806) | | **57.5** | **51.3** | 64.7 | 85.6 | **75.9** | 74.68 | 53.1 |
| GPT-4-Turbo | | 55.0 | 46.1 | 62.6 | 84.4 | 73.4 | 63.71 | 49.3 |
| Claude-3.5-Sonnet | | 52.4 | 40.6 | **68.1** | **88.0** | 71.1 | **76.12** | **59.4** |
| Llama-3.1-8B Instruct | Reported | 20.9 | 21.8 | 45.6 | 79.5 | **45.0** | 40.8 | 23.7 |
| | Reproduced | 34.6 | 33.3 | 39.7 | 76.5 | 24.3 | 37.2 | 45.6 |
| | Vote@16 | — | | | | 26.8 | 43.4 | 56.4 |
| | BoN@16 | 35.8 | 35.1 | 46.5 | 80.7 | 30.8 | 50.5 | 58.8 |
| | PBS@4 | **39.9** | **42.2** | **47.2** | 75.59 | 31.3 | 47.8 | 52.8 |
| Mistral-Nemo Instruct-202407 | Reproduced | 45.0 | 37.5 | 35.7 | 64.2 | 31.9 | 36.6 | 31.4 |
| | Vote@16 | — | | | | 35.2 | 44.4 | 40.7 |
| | BoN@16 | 48.4 | 39.0 | 36.6 | 69.5 | 35.5 | 48.2 | 52.3 |
| | PBS@4 | **53.2** | **49.8** | **42.0** | 63.4 | **37.9** | 44.0 | 47.8 |
| Llama-3.1-70B Instruct | Reported | 38.1 | 29.9 | 55.9 | 85.8 | **65.8** | 55.0 | 41.9 |
| | Reproduced | 42.4 | 41.6 | 59.1 | 85.4 | 45.7 | 55.4 | 63.6 |
| | Vote@16 | — | | | | 51.4 | 60.3 | 70.7 |
| | BoN@16 | **44.4** | **42.9** | **61.9** | **87.7** | 49.8 | **67.1** | **72.0** |

curve. In contrast, our proposed OpenPRM outperforms InternRM, showing promising results in scaling up the best-of-N sampling. However, achieving scaling comparable to the coverage curve remains a substantial challenge. We will release all of these sampling data to the public to encourage further study on process and outcome reward models for inference-time scaling.

# 6 Discussion

## 6.1 Ablation Study of PRM Training

We conducted an ablation study on OpenPRM training, focusing on data sources and model configurations. As illustrated in Table 4a, we compared the effects of continuously fine-tuning InternRM and FsfairX using process pairs built upon preference trees and outcome pairs. The results indicate that process pairs yield superior outcomes, thus validating the effectiveness of our method described in Section 3.1. Additionally, the performance of InternRM, when using a shared prefix, was inferior to configurations using distinct prefixes for chosen and rejected pairs, emphasizing the importance of semantic consistency. Furthermore, while using only UltraFeedback data showed promising results in Chat tasks, maintaining diversity and generalization for open-domain applications is crucial. Therefore, we opted to integrate additional reasoning and STEM questions.

## 6.2 Rewards Shape and Length Bias

We compare the reward shapes of OpenPRM with other reward models in Figure 5. We analyze the rewards of chosen and rejected candidates for each instruction in RewardBench and observe that while all reward models can generally distinguish between chosen and rejected candidates, indicated by a shift in distributions, there remains some overlap. However, OpenPRM exhibits the minimal overlap among them, similar to the parent reward model (i.e., InternRM).

Language model-based judgers often suffer from length bias, typically awarding higher rewards to longer responses. We address this issue in our analysis of OpenPRM, visualizing the correlation between OpenPRM and InterRM in Figure 4b. The results indicate that OpenPRM maintains a correlation of 0.05, compared to 0.37 for InterRM, suggesting that process-level modeling effectively reduces length bias. This also demonstrates the effectiveness and necessity of developing PRMs.

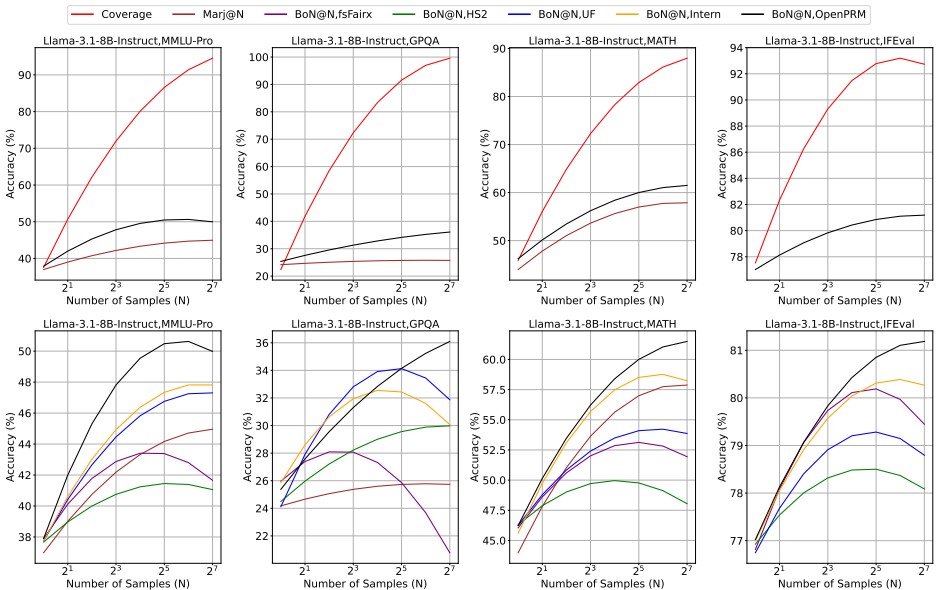

Figure 3: Results of scaling inference-time for Llama-3.1-8B-Instruct on open-domain tasks. These results illustrate the effectiveness of OpenPRMs relative to existing reward models, yet they also highlight the distance to the upper boundary of coverage accuracy (red curve).

| Model | Data | RB Avg. |
|---|---|---|
| InternRM | PrefTree Pairs | 91.1 |
| | w/ Shared Prefix | 90.8 |
| FsfairX | PrefTree Pairs | 89.4 |
| Llama-3-8B-It | PrefTree Pairs | 87.2 |
| InternRM | Outcome Pairs | 88.4 |
| FsfairX | Outcome Pairs | 87.7 |
| Llama-3-8B-It | Outcome Pairs | 86.8 |

(a) Ablation Study of OpenPRM.

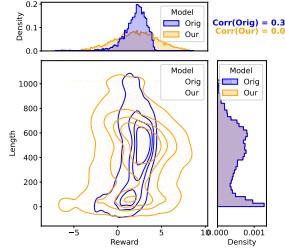

(b) Rewards VS. Length

Figure 4: Ablation Study of OpenPRM

# 7 CONCLUSION

In this paper, we explore the development of process-based reward models (PRMs) in the open domain. We begin by generalizing rewards from outcome-level to process-level, significantly reducing data annotation costs. We then propose the construction of preference trees with parallel candidates for open-domain instructions, from which we derive process pairs using back-propagation. Leveraging this data, our trained OpenPRM achieves excellent results on reward benchmarks and performs well under scaling inference-time search conditions. However, our findings also highlight that there is still considerable progress to be made in building open-domain PRMs to achieve high coverage accuracy. In conclusion, we try to unify ORMs and PRMs in the open domain, paving a new path for PRM development that diverges from domains such as mathematics and coding. We hope that OpenPRM will spark new insights into this topic and stimulate further research.

## ACKNOWLEDGMENTS

This work is supported by the National Science and Technology Major Project (2023ZD0121403), Young Elite Scientists Sponsorship Program by CAST (2023QNRC001), and National Natural Science Foundation of China (No. 62406165).

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

## A  DETAILS ABOUT DERIVATION OF PRM FROM ORM

The cumulative error from supervising only the outcome can be expressed as follows:

$$\text{Cumulative Error} = \sum_{i=1}^{T} \epsilon_i \cdot \frac{1 - \alpha^{T-i+1}}{1 - \alpha} \tag{4}$$

where $\epsilon_i$ is the error at step $i$, and $\alpha$ represents the degree to which errors propagate to subsequent steps.

The reduction in cumulative error can be expressed as:

$$\text{Cumulative Error}^{\text{new}} = \sum_{i=i^*}^{T} \beta \cdot \epsilon_i \cdot \frac{1 - \alpha^{T-i+1}}{1 - \alpha} \tag{5}$$

where $\beta$ is a reduction factor that depends on the discrepancy between $p_c$ and $p_r$, and the strength of the supervision. We define $\beta$ as follows:

$$\beta = \frac{1}{1 + \gamma \cdot \Delta(p_c, p_r) \cdot S} \tag{6}$$

where $\Delta(p_c, p_r)$ is the discrepancy between the key steps, $\gamma$ is a model-dependent factor representing the model's sensitivity to supervision, and $S$ is the strength of the supervision signal. As $\Delta(p_c, p_r)$ and $S$ increase, the factor $\beta$ decreases, leading to a significant reduction in cumulative error.

## B  DETAILS ABOUT OPENPRM TRAINING

### B.1  DATASETS FOR PREFERENCE TREE BUILDING

As introduced in Section 4.1, we construct preference tree data primarily using UltraFeedback, which consists of a mixture of instructions in the open domain. Additionally, we incorporate instructions from the Math and STEM domains to enhance the generalization and reasoning capabilities of open-domain models. Beyond process pairs built with preference trees, we also include some outcome-level preference pairs to maintain the capabilities of ORMs, which can be seen as a specific case of PRMs. We provide all the statistics of the datasets used in Table 4.

| Dataset | Orig | Retain | Outcome? | Process? |
|---|---|---|---|---|
| UltraFeedback | 59,876 | N/A | ✓ | ✗ |
| +PTS | N/A | 70,068 | ✗ | ✓ |
| ScienceQA | 6,508 | N/A | ✓ | ✗ |
| +PTS | N/A | 12,958 | ✗ | ✓ |
| MATH | 7,500 | N/A | ✓ | ✗ |
| +PTS | N/A | 19,913 | ✗ | ✓ |
| HelpSteer 2 | N/A | 7,221 | ✓ | ✗ |

Table 4: Statistics of training datasets. PTS is preference tree sampling strategy proposed in § 3.1.

**Training Data Format.** When preparing the training data for the PRM, we reformat all process-level and outcome-level pairs into a unified format: $[Q, C, P_c, P_r]$, where $P_c$ and $P_l$ represent the chosen (preferred) and rejected (non-preferred) answers, respectively, based on the same context $C$. For outcome-level pairs, $[C, P_c]$ and $[C, P_r]$ represent the complete answers. For process-level pairs, these concatenations represent partial answers.

### B.2  HYPER-PARAMETERS FOR TRAINING

For building preference trees, we compute the edit distance on segments of different response candidates, splitting the entire responses using ".\n". The threshold for segment aggregation is set differently for all tasks based on the distribution of similarity, and the rewards gap threshold between process pairs is 1.0 for UltraFeedback and 0.2 for Math and ScienceQA. For both Llama-3.1-8B-Instruct and Llama-3.1-70B-Instruct, we sample 64 candidate responses for each instruction. Due to

constraints on inference cost, we randomly sample 10,000 instructions from UltraFeedback for the Llama-3.1-70B-Instruct model. We set the temperature to 0.5 and top-p to 1.0 for repeated sampling with vLLM engine [3].

For reproducing the UltraFeedback and HelpSteer2 reward models, we finetune Llama-3-8B-Instruct using a learning rate $5 \times 10^{-6}$ over 1 epoch. Meanwhile, we finetune InternRM and FsfairX on process pairs using a learning rate $1 \times 10^{-6}$ over 1 epoch. All models are finetuned with a batch size of 64 and a maximum sequence length of 2048.

### B.3 TREE BUILDING EXAMPLE

As illustrated in Figure 2 (mainly segment and aggregate), we first split each answer into sentences and then merge similar sentences across all answers sequentially. For each merging operation, the candidate sentences are sourced from the same parent node and indexed consistently across their respective answers. To clarify, we provide a toy example along with the corresponding tree for merging three answers from ScienceQA in Table 5.

Table 5: Example for similarity-based sentence merging for preference building.

**Node(Depth0):** "Question:\nWhich logical fallacy is used in the text?\nBefore I refute my opponent's argument, I would like to draw attention to the fact that he is sweating and clearly does not have much experience on the debate team.\nOptions:\nA. ad hominem: an attack against the person making the argument, rather than the argument itself\nB. bandwagon fallacy: the assumption that the popular choice is automatically correct\nC. appeal to nature: the assumption that natural things are always good\nAnswer: Let's think step by step." — Reward: [] — Reply Indices: []

  **Node(Depth1)**: "To identify the logical fallacy used in the text, let's analyze the situation step by step" — Reward: [0, 1] — Reply Indices: [0, 2]
  Similar sentences: "To identify the logical fallacy used in the text, let's analyze the situation step by step" ——— "To identify the logical fallacy in the text, let's break down the reasoning process step by step"

    **Node(Depth2)**: "The text states that the speaker wants to refute their opponent's argument, but instead of addressing the argument itself, they choose to comment on their opponent's physical appearance (sweating) and a supposed lack of experience on the debate team. This kind of comment is meant to discredit the opponent rather than engaging with the argument they presented" — Reward: [0] — Reply Indices: [0]
      **Node(Depth3)**: "The key phrase here is 'attack against the person making the argument, rather than the argument itself,' which is the definition of the ad hominem fallacy. \n\nTherefore, the correct answer is:\nA. ad hominem: an attack against the person making the argument, rather than the argument itself." — Reward: [0] — Reply Indices: [0]

    **Node(Depth2)**: "The text states, 'Before I refute my opponent́s argument, I would like to draw attention to the fact that he is sweating and clearly does not have much experience on the debate team.' This statement does not address the opponent́s argument itself but instead makes a personal attack on the opponent́s appearance (sweating) and experience, implying that these characteristics make his argument weak" — Reward: [1] — Reply Indices: [2]
      **Node(Depth3)**: "Now, let's analyze the options:\n\nA. ad hominem (A): This type of fallacy involves attacking the person making the argument rather than addressing the argument itself. In this case, the statement attacks the opponent's person (by mentioning their sweating and lack of experience) rather than addressing the merits of their argument" — Reward: [1] — Reply Indices: [2]
        **Node(Depth4)**: "B. bandwagon fallacy (B): This fallacy occurs when it is assumed that something is correct or good because it is popular. There is no indication in the text that the opponent's argument is being rejected because it is unpopular, but rather because of personal characteristics" — Reward: [1] — Reply Indices: [2]

---

[3] https://github.com/vllm-project/vllm

**Node(Depth5)**: "C. appeal to nature (C): This fallacy involves assuming that something is good or true because it is natural. The text does not mention anything about what is natural or unnatural, so this option does not apply" — Reward: [1] — Reply Indices: [2]

**Node(Depth6)**: "Given the analysis, the correct answer is A. ad hominem, as the text attacks the opponent's person rather than addressing the argument itself." — Reward: [1] — Reply Indices: [2]

**Node(Depth1)**: "To identify the logical fallacy in the text, let's break it down step by step:\n\n1. **Identify the claim and the counterargument**: The text claims that the opponent's argument is flawed but before refuting it, it mentions that the opponent is sweating and lacks experience on the debate team. This is an attempt to undermine the opponent's credibility without addressing the argument itself" — Reward: [1] — Reply Indices: [7]

**Node(Depth2)**: "2. **Analyze the nature of the attack**: The attack is not on the argument's merits but on the opponent's character (sweating, lack of experience). Sweating can be a sign of nervousness, which isn't inherently indicative of the validity of one's argument. Lack of experience is also not directly related to the quality of the argument unless it's the first time the opponent is making an argument, which isn't specified" — Reward: [1] — Reply Indices: [7]

**Node(Depth3)**: "3. **Classify the fallacy based on the analysis**: The attack is focused on undermining the opponent personally rather than addressing the argument. This matches the description of the "ad hominem" fallacy, which involves attacking the person making an argument rather than addressing the argument itself" — Reward: [1] — Reply Indices: [7]

**Node(Depth4)**: "Therefore, the logical fallacy used in the text is **ad hominem**" — Reward: [1] — Reply Indices: [7]

**Node(Depth5)**: "The final answer is: A" — Reward: [1] — Reply Indices: [7]

## B.4 RESULTS OF REWARD SHAPE

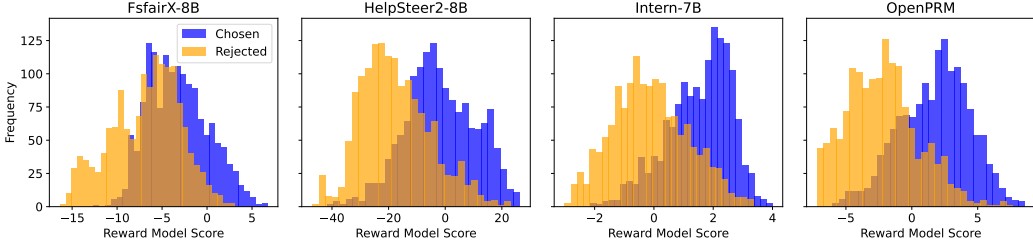

Figure 5: Rewards on chosen and rejected content of various RMs.

## C DETAILS ABOUT EVALUATIONS

We summarize the open-domain datasets used in our evaluations as follows:

- **Alpaca Eval 2** (Dubois et al., 2024): A popular benchmark for evaluating instruction-based language models using automatic evaluators such as GPT-4. It features approximately 800 open-domain prompts. Given the length bias in GPT-4 evaluations, Alpaca Eval 2 employs length-controlled win-rates.

- **MixEval Hard** (Ni et al., 2024): A ground-truth-based dynamic benchmark derived from established benchmark mixtures. It evaluates LLMs using a highly capable model ranking system. MixEval Hard includes both free-form and multiple-choice questions, each category containing 500 questions.

- **MATH500** (Lightman et al., 2023): A subset of the MATH test dataset from OpenAI, featuring 12,500 challenging competition mathematics problems. We use the MATH500 version, which contains 500 samples that maintain IID consistency with the original test dataset, to evaluate the mathematics abilities of LLMs under scaled inference-time settings.

- **MMLU-PRO** (Wang et al., 2024d): An enhanced benchmark designed to evaluate models across broader and more challenging tasks. Built upon the MMLU dataset, MMLU-PRO integrates more challenging, reasoning-focused questions and increases the number of answer choices per question from four to ten, significantly raising the difficulty and reducing the chance of success through random guessing. We randomly sample 500 questions from the test data for our evaluations.

- **GPQA** (Rein et al., 2023): Consists of PhD-level STEM questions generated by experts in biology, physics, and chemistry. The original GPQA dataset is divided into main, diamond, and extended parts. We utilize the diamond split to align with OpenAI results, which includes about 200 questions.

- **IFEval** (Zhou et al., 2023): Designed to evaluate the instruction-following abilities of chat models. It focuses on a set of verifiable instructions and includes over 500 prompts with tasks such as "write an article with more than 800 words" and "wrap your response with double quotation marks."

During implementation, we use zero-shot chain-of-thought promtps for MATH and GPQA datasets based on prompt code in openai `simple-evals` repository [4]. For Alpaca Eval 2, MixEval, IFEval and MMLU-Pro dataset, we use the evaluation code from the official GitHub repository [5][6][7]. Specifically, we found that Mistral-Nemo struggles to adhere to answer format instructions; therefore, we opted to use few-shot Chain of Thought (CoT) examples instead of zero-shot CoT.

## D  ADDITIONAL RESULTS

### D.1  DIFFERENT AGGREGATION STRATEGIES

For applying PRM to outcome-level pairs, we explored several aggregation strategies for calculating step-based rewards, as detailed below:

- Last Step: Use the reward of the final step as the overall reward, similar to ORM.

- Max/Min Step: Select the maximum or minimum reward among all steps.

- Simple Average: Calculate the average reward across all steps.

- Weighted Average: Apply a positional weight, giving later steps higher importance. The formula is: $r = \frac{1}{N} \sum_{i=1}^{N} \frac{i}{N} r_i$.

- Dynamic Aggregation: Utilize uncertainty-weighted optimization (UWO) (Coste et al., 2023), which dynamically adjusts weights based on intra-ensemble variance. This penalizes policies generating outputs with high disagreement across steps.

- Max/Min Delta: Inspired by recent research, we also compute the delta (difference) between step rewards and use the maximum or minimum delta as the final reward.

**Different Performance on Chat Category.** As shown in Table 2 and 7, OpenPRM's scores in the Chat category decrease, while performance in the Chat Hard category improves. This reflects the inherent trade-off between optimizing for simpler conversational tasks (Chat Easy) and more complex reasoning tasks (Chat Hard), which is a known challenge in reward model optimization, as also noted in RewardBench (Lambert et al., 2024). The distinction between these categories lies in their data sources: the Chat category includes datasets like AlpacaEval and MT Bench, while Chat Hard is derived from MT Bench and LLMBar. The decreased scores in the Chat category are largely due to AlpacaEval, as there is a distributional shift between this dataset and our training data, which is sourced from UltraFeedback.

---

[4] https://github.com/openai/simple-evals/
[5] https://github.com/tatsu-lab/alpaca_eval
[6] https://github.com/Psycoy/MixEval
[7] https://github.com/TIGER-AI-Lab/MMLU-Pro

Table 6: Overall score of RewardBench (including Chat, Chat Hard, Safety and Reasoning) and PRM800k Test Results with different aggregation strategies.

| Models | Aggregation | Overall | Chat | Chat Hard | Safety | Reasoning | PRM800k Test |
|---|---|---|---|---|---|---|---|
| intern2-7b-RM | ORM | 87.6 | 99.2 | 69.5 | 87.2 | 94.5 | 61.0 |
| intern2-7b-RM + FT | ORM | 89.6 | 87.7 | 84.2 | 92.4 | 94.1 | 63.4 |
| OpenPRM (intern) | Last Step | 91.1 | 98.0 | 81.6 | 89.5 | 95.1 | 68.1 |
| | Min Step | 91.9 | 96.7 | 83.6 | 91.6 | 95.7 | 68.1 |
| | Max Step | 88.7 | 95.0 | 80.5 | 88.2 | 91.1 | 68.1 |
| | Simple Avg | 91.3 | 97.2 | 83.3 | 89.9 | 94.8 | 68.1 |
| | Weight Avg | 91.4 | 98.0 | 82.7 | 89.7 | 95.0 | 68.1 |
| | Dynamic | 91.6 | 97.8 | 83.3 | 90.4 | 95.0 | 68.1 |
| | Min Delta | 77.8 | 64.3 | 75.7 | 76.4 | 95.1 | 68.1 |
| | Max Delta | 77.7 | 77.4 | 69.5 | 74.3 | 89.5 | 68.1 |

## D.2 EVALUATE RMS WITH OFFSETBIAS

In addition to benchmarks like RewardBench, UltraFeedback, and PRM800k, we evaluated our reward models using OffsetBias (Park et al., 2024), which provides a more granular assessment of bias in reward models. The results are summarized in the table below, showcasing the effectiveness of OpenPRM across various bias metrics in Table 7.

Table 7: OffsetBias Evaluation Results

| Models | Concreteness | Content/ Continuation | Empty Reference | Familiar Knowledge Preference | Length Bias | Nested Instruction | Overall |
|---|---|---|---|---|---|---|---|
| Eurus-RM-7B | 71.4 | 66.7 | 84.6 | 33.3 | 41.2 | 66.7 | 60 |
| FsfairX-LLaMA3-RM | 100 | 91.7 | 53.8 | 91.7 | 41.2 | 58.3 | 71.3 |
| FsfairX-LLaMA3-RM + OffsetBias | 92.9 | 100 | 46.2 | 58.3 | 82.4 | 83.3 | 77.5 |
| LLaMA3-8B-Instruct + OffsetBias | 100 | 95.8 | 92.3 | 83.3 | 85.3 | 50 | 85 |
| Intern2-7b-RM | 1 | 1 | 1 | 58.3 | 58.8 | 91.7 | 84.8 |
| Intern2-7b-RM + OpenPRM (Our) | 92.86 | 83.3 | 1 | 83.3 | 88.2 | 91.7 | 89.9 |

## D.3 COMPARISON WITH MCTS METHODS

We compared the computational efficiency of our method with MCTS-based approaches under similar sampling budgets. The experimental setup and results are as follows:

- OpenPRM: We start from 60k samples, with 64 responses per sample, producing approximately 3.84M question-answer pairs in 24 hours. From this, 90k pairs were selected for training.

- MCTS: We start from 10k samples, with 4 responses per sample. Responses were split into sentences, resulting in 240k partial outputs. Sampling 8 full paths for each pair of partial outputs produced 3.84M question-answer pairs in 24 hours. Of these, 97k pairs were used for training.

The results of PRM trained with OpenPRM and MCTS are shown in Table 8. In fact, the MCTS method remains a powerful baseline but is significantly more resource-intensive, requiring up to 10 times the computational cost of our approach. With a larger sampling budget, the MCTS method could still be effectively utilized. However, our method offers a more efficient alternative and can be further optimized with more accurate similarity computation techniques, such as embedding-based methods. While these approaches may increase the time required to construct the tree, they hold great potential for improving performance. We plan to explore these optimizations in future work to further enhance the efficiency and accuracy of our method.

Table 8: Comparison with MCTS methods

| Models | Aggregation | Overall | Chat | Chat Hard | Safety | Reasoning | PRM800k Test |
|---|---|---|---|---|---|---|---|
| intern2-7b-RM | ORM | 87.6 | 99.2 | 69.5 | 87.2 | 94.5 | 61 |
| intern2-7b-RM + outcome labels | ORM | 89.6 | 87.71 | 84.21 | 92.43 | 94.06 | 63.4 |
| OpenPRM (intern) | Min Step | 91.9 | 96.7 | 83.6 | 91.6 | 95.7 | 68.1 |
| MCTS (intern) | Min Step | 91.4 | 95.5 | 81.6 | 93.2 | 95.4 | 68.2 |

## D.4 RESULTS ON MORE LANGUAGE MODELS

**Scaling Effect of PBS.** We have analyzed the scaling effects of Process Beam Search (PBS), where beam search is conducted at the sentence level, selecting the top N generated outputs based on PRM rewards. As presented in Figure 6, the results indicate that OpenPRM consistently outperforms Bag of N-grams (BoN) in PBS settings, showcasing its effectiveness and reliability in these scenarios. However, we also observed significant variance in the scaling effect of PBS across different tasks, in contrast to the more consistent scaling effect seen with best-of-N methods. This variance can likely be attributed to differences in the data distributions across tasks, which highlight the need for further investigation into handling data mixtures effectively. We plan to continue exploring this issue in future work to better understand and address these challenges.

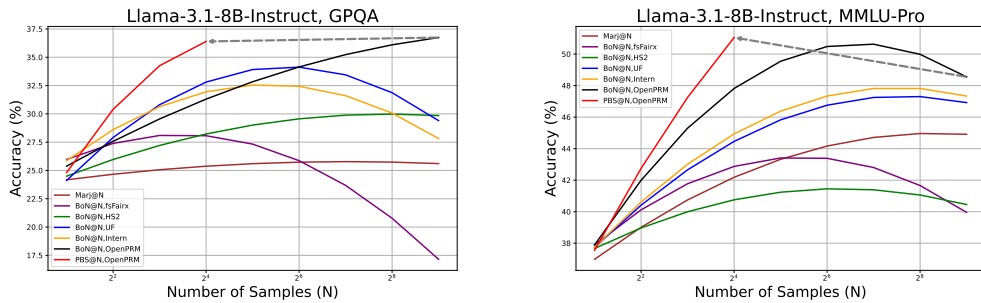

Figure 6: Scaling Effect of Process Beam Search

**Scaling Effect of BoN.** We present the remaining tasks, in addition to those shown in Figure 3, in Figure 7. We provide additional results for the Llama-3.1-70B-Instruct and Mistral-Nemo model regarding inference-time scaling in Figures 8, 9, 10 and 11, which support the same conclusions as those drawn from the Llama-3.1-8B-Instruct in Figure 3.

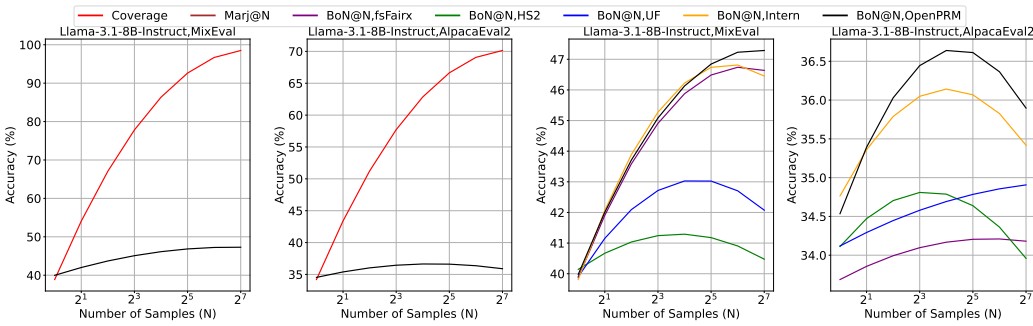

Figure 7: Results of scaling inference-time for Llama-3.1-8B-Instruct on the rest tasks of Figure 3.

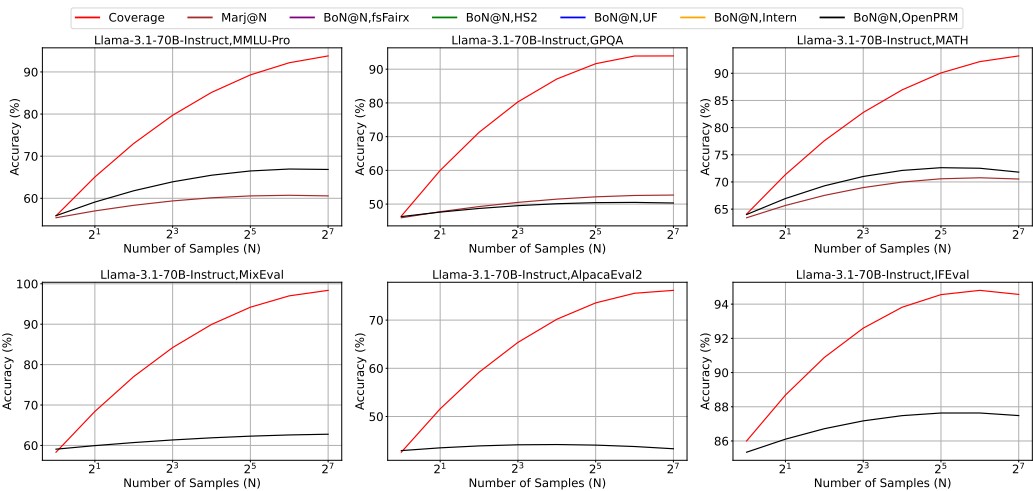

Figure 8: Results of scaling inference-time for Llama-3.1-70B-Instruct on open-domain tasks.

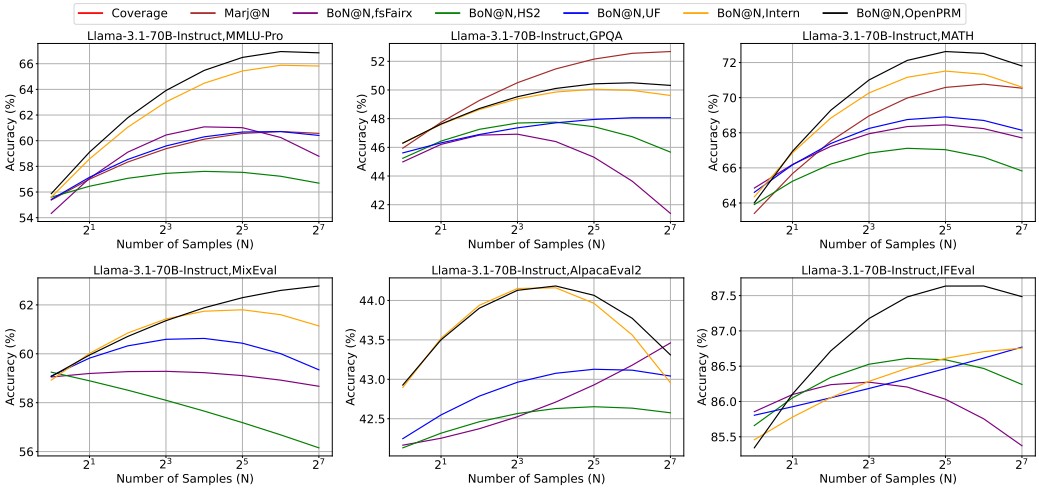

Figure 9: Results of scaling inference-time for Llama-3.1-70B-Instruct on open-domain tasks.

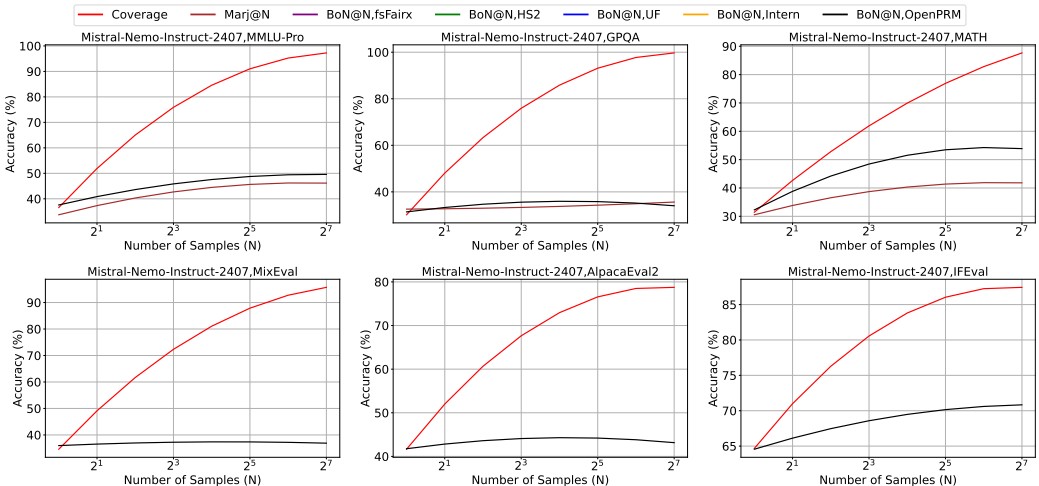

Figure 10: Results of scaling inference-time for Mistral-Nemo on open-domain tasks.

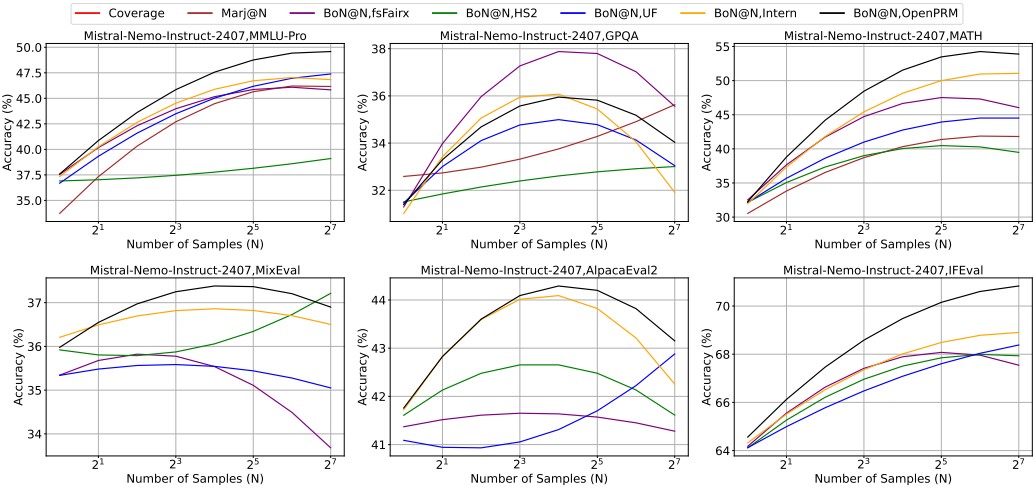

Figure 11: Results of scaling inference-time for Mistral-Nemo on open-domain tasks.

