# OpenReview forum: "OpenPRM: Building Open-domain Process-based Reward Models with Preference Trees"
_ICLR.cc/2025/Conference — ICLR 2025 Poster_

### Official Review · Reviewer_MtX7 · 2024-11-02

**Soundness:** 3
**Presentation:** 3
**Contribution:** 3
**Rating:** 6
**Confidence:** 4

**Summary:**

This paper introduces a sentence-level process reward data collection method using Monte Carlo Tree Search (MCTS). The authors use outcome-based rewards and iteratively sample, segment outputs, aggregate at nodes, and apply back-propagation. Experimental results demonstrate that the collected process reward data effectively trains the process reward model.

**Strengths:**

Clear and well-structured writing makes the methodology easy to follow.
The method addresses the popular and relevant area of automatic process reward data collection.
Effective use of MCTS enhances the quality of data for training process reward models.

**Weaknesses:**

Compared to previous MCTS-based solutions for process reward models, the novelty of the data creation method is unclear. Further explanation from the authors on this point would be beneficial.

## Update after author response
The authors conducted additional experiments to clarify the contributions of OpenPRM compared to MCTS. From the rebuttal results, it is evident that MCTS remains a strong baseline in terms of performance. While the authors claim that OpenPRM is 10x more efficient than MCTS, they also state that both methods can process 3.84M examples within 24 hours. This apparent contradiction requires further clarification from the authors. Overall, OpenPRM's novelty appears limited, positioning it as an incremental improvement.

The authors also provided comparisons between ORM and PRM, offering additional insights into the value of PRM.

Based on the new evidence, I increase my score to 6, indicating a borderline accept.

**Questions:**

In Table 2, could the authors clarify how comparisons are made across different backbones and training data to ensure an "apple-to-apple" comparison?

Regarding Table 3, could the authors provide the results from the outcome reward model?

---

> ### Author Response · Authors · 2024-11-23
> **Response - 1**
>
> We sincerely thank you for your positive feedback and valuable suggestions. Below, we provide detailed responses to address your concerns and clarify our approach.
>
> ### Q1: Comparsion with MCTS-based solutions
>
> We compared the computational efficiency of our method with MCTS-based approaches under similar sampling budgets. The experimental setup and results are as follows:
>
> **A1. Sampling Settings:**
> -  **OpenPRM:** We start from 60k samples, with 64 responses per sample, producing approximately 3.84M question-answer pairs in 24 hours. From this, 90k pairs were selected for training.
> - **MCTS:** We start from 10k samples, with 4 responses per sample. Responses were split into sentences, resulting in 240k partial outputs. Sampling 8 full paths for each pair of partial outputs produced ~3.84M question-answer pairs in 24 hours. Of these, 97k pairs were used for training.
>
> **A2. Results:**
> The results of PRM trained with OpenPRM and MCTS are shown below:
>
> | Models                             | Aggregation | RewardBench |       |           |        |           | PRM800k Test |
> | ---------------------------------- | ----------- | ----------- | ----- | --------- | ------ | --------- | ------------ |
> |                                    |             | Overall     | Chat  | Chat Hard | Safety | Reasoning |              |
> | intern2-7b-RM                      | ORM         | 87.6        | 99.2  | 69.5      | 87.2   | 94.5      | 61           |
> | intern2-7b-RM<br/>+ outcome labels | ORM         | 89.6        | 87.71 | 84.21     | 92.43  | 94.06     | 63.4         |
> | OpenPRM (intern)                   | Min Step    | 91.9        | 96.7  | 83.6      | 91.6   | 95.7      | 68.1         |
> | MCTS (intern)                      | Min Step    | 91.4        | 95.5  | 81.6      | 93.2   | 95.4      | 68.2         |
>
> In fact, the MCTS method remains a powerful baseline but is significantly more resource-intensive, requiring up to 10 times the computational cost of our approach. With a larger sampling budget, the MCTS method could still be effectively utilized. However, our method offers a more efficient alternative and can be further optimized with more accurate similarity computation techniques, such as embedding-based methods. While these approaches may increase the time required to construct the tree, they hold great potential for improving performance.
>
> We plan to explore these optimizations in future work to further enhance the efficiency and accuracy of our method.
>
>
> ### Q2: Clarification of Comparisons in Table 2
>
> We acknowledge that this is an important issue that warrants careful consideration. The use of different datasets for different models introduces challenges in achieving fully comparable ablations.
> To address this, we considered the backbone models (Intern-RM and FsFairX-RM) as baselines and further fine-tuned these models using datasets containing both outcome-level and process-level labels. For the ablation study, the dataset was split into two types: **Preference Tree Pairs** (process-level) and **Outcome Pairs** (outcome-level).
>
> As shown in Figure 5(a), our results demonstrate that OpenPRM, trained on process-level labels (Preference Tree Pairs), outperforms models fine-tuned solely on outcome-level labels. This finding underscores the effectiveness of incorporating process-level training for enhancing performance.

---

> ### Author Response · Authors · 2024-11-23
> **Response - 2**
>
> ### Q3: Additional Results from ORM in Table 3
>
> In Figure 3, we compared various outcome-level reward models (ORMs) to evaluate their performance when scaling best-of-N selections. The results highlight the superior performance of OpenPRM. To improve readability, we will include some of these results in Table 3 for better reference.
> Regarding process-level beam search (PBS), scoring individual steps requires a reward model capable of evaluating partial outputs. Unfortunately, current outcome-level reward models are not designed for this purpose, which limits the comparison. Consequently, we have only included results for OpenPRMs, as no other open-domain PRMs are available for direct comparison.
>
> We have analyzed the scaling effects of Process Beam Search (PBS), where beam search is conducted at the sentence level, selecting the top N generated outputs based on PRM rewards.
> As presented in the following table, the results indicate that OpenPRM consistently outperforms Bag of N-grams (BoN) in PBS settings, showcasing its effectiveness and reliability in these scenarios.
>
> | Models     | Setting       | GPQA  |       |       |       |       |       |       |       | MMLU-Pro |       |       |       |       |       |       |       |
> | ---------- | ------------- | ----- | ----- | ----- | ----- | ----- | ----- | ----- | ----- | -------- | ----- | ----- | ----- | ----- | ----- | ----- | ----- |
> |            |               | 1     | 2     | 4     | 8     | 16    | 32    | 64    | 128   | 1        | 2     | 4     | 8     | 16    | 32    | 64    | 128   |
> | -          | Coverage      | 24.34 | 39.39 | 57.58 | 72.32 | 84.95 | 92.12 | 96.87 | 98.99 | 37.20    | 51.24 | 62.12 | 72.00 | 79.80 | 86.56 | 91.44 | 94.6  |
> | -          | Majority Vote | 24.34 | 25.25 | 23.03 | 26.16 | 26.77 | 25.56 | 24.95 | 26.06 | 37.20    | 38.40 | 41.28 | 42.00 | 43.44 | 44.12 | 44.64 | 45    |
> | FsFairX-RM | ORM - BoN     | 24.34 | 28.79 | 29.39 | 28.69 | 26.77 | 24.04 | 23.03 | 22.12 | 37.20    | 41.00 | 42.16 | 42.44 | 43.52 | 43.24 | 42.16 | 42.2  |
> | Intern-RM  | ORM - BoN     | 24.34 | 30.1  | 31.72 | 32.32 | 31.92 | 30.81 | 31.52 | 27.98 | 37.2     | 42.64 | 45.36 | 46.56 | 48.12 | 48.52 | 48.88 | 48.48 |
> | OpenPRM    | PRM - BoN     | 24.34 | 28.59 | 30    | 31.92 | 32.63 | 33.13 | 34.34 | 37.17 | 37.20    | 42.84 | 46.04 | 46.88 | 49.56 | 50.52 | 50.32 | 50.28 |
> | OpenPRM    | PRM - PBS     | 24.34 | 31.82 | 32.83 | 36.87 | -     | -     | -     | -     | 37.20    | 43.8  | 46.2  | 51.4  | -     | -     | -     | -     |
>
> However, we also observed significant variance in the scaling effect of PBS across different tasks, in contrast to the more consistent scaling effect seen with best-of-N methods. This variance can likely be attributed to differences in the data distributions across tasks, which highlight the need for further investigation into handling data mixtures effectively.
> We plan to continue exploring this issue in future work to better understand and address these challenges.
>
> ---
>
> We welcome further discussion on this topic and appreciate your interest in our results.

---

> > ### Comment · Reviewer_MtX7 · 2024-11-24
> > **See weakness for review updates**
> >
> > I have increased my score based on the new results presented in the author response. See the weaknesses section for detailed review updates.

---

### Official Review · Reviewer_4T3D · 2024-11-03

**Soundness:** 2
**Presentation:** 2
**Contribution:** 2
**Rating:** 5
**Confidence:** 4

**Summary:**

This paper presents OpenPRM, a novel approach to developing process-based reward models (PRMs) for open-domain tasks like instruction following. The authors argue that while outcome-based reward models (ORMs) have shown success, their coarse-grained nature limits their effectiveness in supervising complex tasks. OpenPRM leverages existing ORMs and preference trees based on parallel sampled candidates to generate process-level rewards, leading to improved performance on various benchmarks. The paper also explores scaling inference-time compute in open-domain settings and finds OpenPRM superior to traditional ORMs in scaled settings.

The fundamental of this paper is about the generation of PRM training data, and it is using the similar method as Math-Shepherd and OmegaPRM. The key innovation of the this paper lies in the idea of merging solutions from the same question into a tree structure based on the similarity of the steps. Given this method would greatly reduce of the cost of the tree, the write and the experiments are not discussed throughly.

**Strengths:**

1. The paper addresses a significant gap in the research by extending the application of PRMs from specialized domains like mathematics to open-domain tasks. The proposed preference tree construction using readily available ORMs is a creative and cost-effective solution to generate process-level supervision.
2. OpenPRM consistently outperforms existing ORMs on RewardBench, particularly in challenging tasks like Chat Hard and Reasoning. Furthermore, it surpasses specialized PRMs in open-domain benchmarks, highlighting its generalizability. OpenPRM also shows promising results in scaling inference-time compute, outperforming other models in best-of-N sampling.
3. The paper provides a detailed analysis of the limitations of ORMs in process-level supervision, highlighting the issue of cumulative error. The authors theoretically and empirically demonstrate how OpenPRM mitigates this problem by identifying key divergent steps in the process. The paper also addresses potential concerns like rationality of process aggregation and length bias, strengthening the validity of the proposed approach.

**Weaknesses:**

1. The tree building in OpenPRM is not convincing. The steps are separated by “.\n”, and the similarity of steps are based on the edit distance between them, and threshold is pre-defined 0.88. The solutions are sampled with temperature 0.5. This temperature may not be high enough to generate solutions that are diverse enough for the reward calculation. I interpret this temperature is needed to generate steps that similar enough to convert into tree, but it affects the diverse of the solution and the rewards may not be precise. And how this 0.88 threshold selected was not mentioned in the paper, and the affect of different thresholds is not discussed either. As the key innovation of this paper, this is not a solid foundation.
2. How the PRM is trained given the data collected was not discussed clearly in the paper. Some information about the mixture and hyper parameters are given, but how the data was structured into training examples, how the rewards are predicted during inference and how the rewards for steps are aggregated into final reward are not discussed. Without these information, it is hard to have a good picture of the PRM training and how to interpret the results presented in the paper.
3. Only one example was given in the Appendix, and it was not clear. I would be very interested to see an real example of how the original solutions are like and how the final tree built looks like.

**Questions:**

Questions would be around the points in Weaknesses section.

1. Why pick the temperature of 0.5 as most of the time, people would use high temperate to get diverse solutions.
2. How the threshold 0.88 for similarity is picked and how does it affect the data and final result?
3. How the training data is formatted and what loss was used? As there are different way of PRM training available, which one was used?
4. During the inference time, how the rewards for each step/solution is calculated and aggregated?
5. Can you give more detailed example of the solution and tree building as it is the core contribution of the paper?

---

> ### Author Response · Authors · 2024-11-23
> **Response - 1**
>
> We sincerely thank you for your positive feedback and valuable suggestions. Below, we provide detailed responses to address your concerns and clarify our approach.
> ### Q1: Details on Tree Building for Training Data
>
> **A1: Temperature:** Thank you for highlighting this issue. The temperature parameter indeed affects the diversity of generated responses. While higher diversity provides a more accurate estimation, it may also reduce similarity between spans, leading to fewer final pairs during merging. To address this, we employ a similarity-based method to merge different answer spans. Consequently, we choose a lower temperature to sample responses, ensuring a better balance between diversity and similarity.
>
> **A2. Edit Distance Analysis:** Following prior studies, we determined that an edit distance above 0.8 (normalized to $[0, 1]$, where 1 indicates complete similarity) is preferable for sentence similarity. The distribution of edit distances for all sentence pairs follows a normal curve, centered around 0.88. Balancing similarity with dataset size, we chose 0.88 as our threshold. The statistical distribution of edit distances in UltraFeedback is presented below.
>
> | 0.78  | 0.80  | 0.82   | 0.84   | 0.86    | 0.88    | 0.90   | 0.92   | 0.94  | 0.96  | 0.98  | 1.00  |
> | ----- | ----- | ------ | ------ | ------- | ------- | ------ | ------ | ----- | ----- | ----- | ----- |
> | 31008 | 72028 | 201067 | 655930 | 1401751 | 1522812 | 569087 | 139557 | 60747 | 34051 | 23702 | 38944 |
>
>
> ### Q2: Training Details of PRM
>
> When preparing the training data for the PRM, we reformat all process-level and outcome-level pairs into a unified format: $(Q, C, P_w, P_l)$, where $P_w$ and $P_l$ represent the chosen (preferred) and rejected (non-preferred) answers, respectively, based on the same context $C$. For outcome-level pairs, $[C, P_w]$ and $[C, P_l]$ represent the complete answers. For process-level pairs, these concatenations represent partial answers.
>
> Using this unified format, we train the PRM using the Bradley-Terry objective as follows:
> $$\mathcal{L}{\text{PRM}}(\theta) = -\frac{1}{N} \sum{i=1}^{N} \log \left( \sigma \left( r_\theta(q_i, c_i, p^i_w) - r_\theta(q_i, c_i, p^i_l) \right) \right)$$
>
> This formulation ensures consistent training across both process- and outcome-level dataset.

---

> > ### Comment · Reviewer_4T3D · 2024-11-26
> > **Thanks for the explanation**
> >
> > The answers align with my assumption, will keep the rating.

---

> ### Author Response · Authors · 2024-11-23
> **Response - 2**
>
> ### Q3: Inference Details of PRM
>
> For applying PRM to outcome-level pairs, we explored several aggregation strategies for calculating step-based rewards, as detailed below:
> - **Last Step:** Use the reward of the final step as the overall reward, similar to ORM.
> - **Max/Min Step:** Select the maximum or minimum reward among all steps.
> - **Simple Average:** Calculate the average reward across all steps.
> - **Weighted Average:** Apply a positional weight, giving later steps higher importance. The formula is: $r = \frac{1}{N} \sum_{i=1}^N \frac{i}{N} r_i$
> - **Dynamic Aggregation:** Utilize uncertainty-weighted optimization (UWO) [1], which dynamically adjusts weights based on intra-ensemble variance. This penalizes policies generating outputs with high disagreement across steps.
> - **Max/Min Delta:** Inspired by recent research, we also compute the delta (difference) between step rewards and use the maximum or minimum delta as the final reward.
>
> | Models                             | Aggregation | RewardBench |       |           |        |           | PRM800k Test |
> | ---------------------------------- | ----------- | ----------- | ----- | --------- | ------ | --------- | ------------ |
> |                                    |             | Overall     | Chat  | Chat Hard | Safety | Reasoning |              |
> | intern2-7b-RM                      | ORM         | 87.6        | 99.2  | 69.5      | 87.2   | 94.5      | 61           |
> | intern2-7b-RM<br/>+ outcome labels | ORM         | 89.6        | 87.71 | 84.21     | 92.43  | 94.06     | 63.4         |
> | OpenPRM<br/>(intern)               | Last Step   | 91.1        | 98.0  | 81.6      | 89.5   | 95.1      | 68.1         |
> |                                    | Min Step    | 91.9        | 96.7  | 83.6      | 91.6   | 95.7      | 68.1         |
> |                                    | Max Step    | 88.7        | 95.0  | 80.5      | 88.2   | 91.1      | 68.1         |
> |                                    | Simple Avg  | 91.3        | 97.2  | 83.3      | 89.9   | 94.8      | 68.1         |
> |                                    | Weight Avg  | 91.4        | 98.0  | 82.7      | 89.7   | 95.0      | 68.1         |
> |                                    | Dynamic     | 91.6        | 97.8  | 83.3      | 90.4   | 95.0      | 68.1         |
> |                                    | Min Delta   | 77.8        | 64.3  | 75.7      | 76.4   | 95.1      | 68.1         |
> |                                    | Max Delta   | 77.7        | 77.4  | 69.5      | 74.3   | 89.5      | 68.1         |
>
> [1] Reward Model Ensembles Help Mitigate Overoptimization. ICLR 2024
> [2] Rewarding Progress: Scaling Automated Process Verifiers for LLM Reasoning. Arxiv 2024
>
>
> ### Q4: Example of Tree Building with Original Solutions
>
> As illustrated in Figure 2 (segment and aggregate), we first split each answer into sentences and then merge similar sentences across all answers sequentially. For each merging operation, the candidate sentences are sourced from the same parent node and indexed consistently across their respective answers. To clarify, we provide a toy example along with the corresponding tree for merging three answers from ScienceQA.

---

> ### Author Response · Authors · 2024-11-23
> **Response - 3**
>
> ```
> Node(Depth0): "Question:\nWhich logical fallacy is used in the text?\nBefore I refute my opponent's argument, I would like to draw attention to the fact that he is sweating and clearly does not have much experience on the debate team.\nOptions:\nA. ad hominem: an attack against the person making the argument, rather than the argument itself\nB. bandwagon fallacy: the assumption that the popular choice is automatically correct\nC. appeal to nature: the assumption that natural things are always good\nAnswer: Let's think step by step." | Reward: [] | Reply Indices: []
>
> 	Node(Depth1): "To identify the logical fallacy used in the text, let's analyze the situation step by step" | Reward: [0, 1] | Reply Indices: [0, 2]
> 	Similar sentences: "To identify the logical fallacy used in the text, let's analyze the situation step by step" ||| "To identify the logical fallacy in the text, let's break down the reasoning process step by step"
>
> 		Node(Depth2): "The text states that the speaker wants to refute their opponent's argument, but instead of addressing the argument itself, they choose to comment on their opponent's physical appearance (sweating) and a supposed lack of experience on the debate team. This kind of comment is meant to discredit the opponent rather than engaging with the argument they presented" | Reward: [0] | Reply Indices: [0]
> 			Node(Depth3): 'The key phrase here is "attack against the person making the argument, rather than the argument itself," which is the definition of the ad hominem fallacy. \n\nTherefore, the correct answer is:\nA. ad hominem: an attack against the person making the argument, rather than the argument itself.' | Reward: [0] | Reply Indices: [0]
>
> 		Node(Depth2): 'The text states, "Before I refute my opponent\'s argument, I would like to draw attention to the fact that he is sweating and clearly does not have much experience on the debate team." This statement does not address the opponent\'s argument itself but instead makes a personal attack on the opponent\'s appearance (sweating) and experience, implying that these characteristics make his argument weak' | Reward: [1] | Reply Indices: [2]
> 			Node(Depth3): "Now, let's analyze the options:\n\nA. ad hominem (A): This type of fallacy involves attacking the person making the argument rather than addressing the argument itself. In this case, the statement attacks the opponent's person (by mentioning their sweating and lack of experience) rather than addressing the merits of their argument" | Reward: [1] | Reply Indices: [2]
> 				Node(Depth4): "B. bandwagon fallacy (B): This fallacy occurs when it is assumed that something is correct or good because it is popular. There is no indication in the text that the opponent's argument is being rejected because it is unpopular, but rather because of personal characteristics" | Reward: [1] | Reply Indices: [2]
> 					Node(Depth5): 'C. appeal to nature (C): This fallacy involves assuming that something is good or true because it is natural. The text does not mention anything about what is natural or unnatural, so this option does not apply' | Reward: [1] | Reply Indices: [2]
> 						Node(Depth6): "Given the analysis, the correct answer is A. ad hominem, as the text attacks the opponent's person rather than addressing the argument itself." | Reward: [1] | Reply Indices: [2]
>
> 	Node(Depth1): "To identify the logical fallacy in the text, let's break it down step by step:\n\n1. **Identify the claim and the counterargument**: The text claims that the opponent's argument is flawed but before refuting it, it mentions that the opponent is sweating and lacks experience on the debate team. This is an attempt to undermine the opponent's credibility without addressing the argument itself" | Reward: [1] | Reply Indices: [7]
> 		Node(Depth2): "2. **Analyze the nature of the attack**: The attack is not on the argument's merits but on the opponent's character (sweating, lack of experience). Sweating can be a sign of nervousness, which isn't inherently indicative of the validity of one's argument. Lack of experience is also not directly related to the quality of the argument unless it's the first time the opponent is making an argument, which isn't specified" | Reward: [1] | Reply Indices: [7]
> 			Node(Depth3): '3. **Classify the fallacy based on the analysis**: The attack is focused on undermining the opponent personally rather than addressing the argument. This matches the description of the "ad hominem" fallacy, which involves attacking the person making an argument rather than addressing the argument itself' | Reward: [1] | Reply Indices: [7]
> 				Node(Depth4): 'Therefore, the logical fallacy used in the text is **ad hominem**' | Reward: [1] | Reply Indices: [7]
> 					Node(Depth5): 'The final answer is: A' | Reward: [1] | Reply Indices: [7]
> ```
>
> We hope these responses address your concerns and provide further clarity. Thank you once again for your constructive feedback.

---

### Official Review · Reviewer_SVaF · 2024-11-05

**Soundness:** 3
**Presentation:** 2
**Contribution:** 3
**Rating:** 5
**Confidence:** 5

**Summary:**

The authors proposed a method to create tree-like process supervision data. Specifically, it samples a great amount of trajectories from a set of samplers (LLMs) and merged those with similar prefix (measured by edit distance) to construct a tree. They trained different PRMs using this data and conducted experiments of Best-of-N and beam-search. The PRM-assisted method performed better on some of the selected benchmarks.

**Strengths:**

1. Process supervision and PRM are very important topics in the field. Especially given the improving importance of LLM reasoning and the success of OpenAI o1 model. This is an emergent topic and an open-accessible dataset is valuable to academia.
2. The inference-time scaling experiment is very valuable. Experimenting with scaling law is always a very important thing in today's LLM research. Although it's often infeasible for most academic labs due to the resource constraints. I am very happy to see such analysis in the papers nowadays. The results are not very convincing though but experimenting it should be awarded with credits.

**Weaknesses:**

1. The biggest issue is the presentation of the paper. Seems many details are missing. For example: how the data merge exactly work. The paper only presented the general idea but I cannot find any detailed description, e.g., the edit distance threshold. Another example is table 2, I cannot find what metrics the table used. Is that per-step accuracy? If so, how could you evaluate PRMs using RewardBench (that's for ORM I remember?)
2. They are some potential errors in the review of previous method. For example, I don't think any of the reward models are trained with MSE loss. It doesn't make sense to train a reward model with 0~1 labels where the label represented a probability using MSE loss. This is not a regression problem. And the Step-DPO work didn't train a reward model at all I believe.

**Questions:**

You mentioned one of the motivation of the proposed method is to save compute. But I don't don't think there are compute difference for crafting a process supervision label. You still need to same amount of compute to sampling the trajectories. Please correct me if I didn't understand correctly here.

---

> ### Author Response · Authors · 2024-11-23
> **Response - 1**
>
> We sincerely thank you for your positive feedback and valuable suggestions. Below, we provide detailed responses to address your concerns and clarify our approach.
>
> ### Q1: Details on Building the OpenPRM Dataset
> We constructed the OpenPRM dataset following these steps:
>
>
> **A1. Edit Distance Analysis:** Following prior studies, we determined that an edit distance above 0.8 (normalized to $[0, 1]$, where 1 indicates complete similarity) is preferable for sentence similarity. The distribution of edit distances for all sentence pairs follows a normal curve, centered around 0.88. Balancing similarity with dataset size, we chose 0.88 as our threshold. The statistical distribution of edit distances in UltraFeedback is presented below.
>
> | 0.78  | 0.80  | 0.82   | 0.84   | 0.86    | 0.88    | 0.90   | 0.92   | 0.94  | 0.96  | 0.98  | 1.00  |
> | ----- | ----- | ------ | ------ | ------- | ------- | ------ | ------ | ----- | ----- | ----- | ----- |
> | 31008 | 72028 | 201067 | 655930 | 1401751 | 1522812 | 569087 | 139557 | 60747 | 34051 | 23702 | 38944 |
>
> **A2. Data Merging:** As illustrated in Figure 2 (segment and aggregate), we first split each answer into sentences and then merge similar sentences across all answers sequentially. For each merging operation, the candidate sentences are sourced from the same parent node and indexed consistently across their respective answers. To clarify, we provide a toy example along with the corresponding tree for merging three answers from ScienceQA.

---

> ### Author Response · Authors · 2024-11-23
> **Response - 2**
>
> ```
> Node(Depth0): "Question:\nWhich logical fallacy is used in the text?\nBefore I refute my opponent's argument, I would like to draw attention to the fact that he is sweating and clearly does not have much experience on the debate team.\nOptions:\nA. ad hominem: an attack against the person making the argument, rather than the argument itself\nB. bandwagon fallacy: the assumption that the popular choice is automatically correct\nC. appeal to nature: the assumption that natural things are always good\nAnswer: Let's think step by step." | Reward: [] | Reply Indices: []
>
> 	Node(Depth1): "To identify the logical fallacy used in the text, let's analyze the situation step by step" | Reward: [0, 1] | Reply Indices: [0, 2]
> 	Similar sentences: "To identify the logical fallacy used in the text, let's analyze the situation step by step" ||| "To identify the logical fallacy in the text, let's break down the reasoning process step by step"
>
> 		Node(Depth2): "The text states that the speaker wants to refute their opponent's argument, but instead of addressing the argument itself, they choose to comment on their opponent's physical appearance (sweating) and a supposed lack of experience on the debate team. This kind of comment is meant to discredit the opponent rather than engaging with the argument they presented" | Reward: [0] | Reply Indices: [0]
> 			Node(Depth3): 'The key phrase here is "attack against the person making the argument, rather than the argument itself," which is the definition of the ad hominem fallacy. \n\nTherefore, the correct answer is:\nA. ad hominem: an attack against the person making the argument, rather than the argument itself.' | Reward: [0] | Reply Indices: [0]
>
> 		Node(Depth2): 'The text states, "Before I refute my opponent\'s argument, I would like to draw attention to the fact that he is sweating and clearly does not have much experience on the debate team." This statement does not address the opponent\'s argument itself but instead makes a personal attack on the opponent\'s appearance (sweating) and experience, implying that these characteristics make his argument weak' | Reward: [1] | Reply Indices: [2]
> 			Node(Depth3): "Now, let's analyze the options:\n\nA. ad hominem (A): This type of fallacy involves attacking the person making the argument rather than addressing the argument itself. In this case, the statement attacks the opponent's person (by mentioning their sweating and lack of experience) rather than addressing the merits of their argument" | Reward: [1] | Reply Indices: [2]
> 				Node(Depth4): "B. bandwagon fallacy (B): This fallacy occurs when it is assumed that something is correct or good because it is popular. There is no indication in the text that the opponent's argument is being rejected because it is unpopular, but rather because of personal characteristics" | Reward: [1] | Reply Indices: [2]
> 					Node(Depth5): 'C. appeal to nature (C): This fallacy involves assuming that something is good or true because it is natural. The text does not mention anything about what is natural or unnatural, so this option does not apply' | Reward: [1] | Reply Indices: [2]
> 						Node(Depth6): "Given the analysis, the correct answer is A. ad hominem, as the text attacks the opponent's person rather than addressing the argument itself." | Reward: [1] | Reply Indices: [2]
>
> 	Node(Depth1): "To identify the logical fallacy in the text, let's break it down step by step:\n\n1. **Identify the claim and the counterargument**: The text claims that the opponent's argument is flawed but before refuting it, it mentions that the opponent is sweating and lacks experience on the debate team. This is an attempt to undermine the opponent's credibility without addressing the argument itself" | Reward: [1] | Reply Indices: [7]
> 		Node(Depth2): "2. **Analyze the nature of the attack**: The attack is not on the argument's merits but on the opponent's character (sweating, lack of experience). Sweating can be a sign of nervousness, which isn't inherently indicative of the validity of one's argument. Lack of experience is also not directly related to the quality of the argument unless it's the first time the opponent is making an argument, which isn't specified" | Reward: [1] | Reply Indices: [7]
> 			Node(Depth3): '3. **Classify the fallacy based on the analysis**: The attack is focused on undermining the opponent personally rather than addressing the argument. This matches the description of the "ad hominem" fallacy, which involves attacking the person making an argument rather than addressing the argument itself' | Reward: [1] | Reply Indices: [7]
> 				Node(Depth4): 'Therefore, the logical fallacy used in the text is **ad hominem**' | Reward: [1] | Reply Indices: [7]
> 					Node(Depth5): 'The final answer is: A' | Reward: [1] | Reply Indices: [7]
> ```

---

> ### Author Response · Authors · 2024-11-23
> **Response - 3**
>
> **A3. Evaluation Metrics:** For outcome-level reward benchmarks, such as RewardBench and UltraFeedback, we compute outcome-level accuracy. For process-level reward benchmarks like PRM800k, we construct a test set with samples in the form $(Q, C, P_1, P_2)$, where $Q$ is the question, $C$ represents the shared context, and $P_1$ and $P_2$ are the target steps. Similar to outcome-level benchmarks, we evaluate accuracy at this level.
>
> **A4. PRM Aggregation Strategies:** For applying PRM to outcome-level pairs, we explored several aggregation strategies for calculating step-based rewards, as detailed below:
> - **Last Step:** Use the reward of the final step as the overall reward, similar to ORM.
> - **Max/Min Step:** Select the maximum or minimum reward among all steps.
> - **Simple Average:** Calculate the average reward across all steps.
> - **Weighted Average:** Apply a positional weight, giving later steps higher importance. The formula is: $r = \frac{1}{N} \sum_{i=1}^N \frac{i}{N} r_i$
> - **Dynamic Aggregation:** Utilize uncertainty-weighted optimization (UWO) [1], which dynamically adjusts weights based on intra-ensemble variance. This penalizes policies generating outputs with high disagreement across steps.
> - **Max/Min Delta:** Inspired by recent research, we also compute the delta (difference) between step rewards and use the maximum or minimum delta as the final reward.
>
> | Models                             | Aggregation | RewardBench |       |           |        |           | PRM800k Test |
> | ---------------------------------- | ----------- | ----------- | ----- | --------- | ------ | --------- | ------------ |
> |                                    |             | Overall     | Chat  | Chat Hard | Safety | Reasoning |              |
> | intern2-7b-RM                      | ORM         | 87.6        | 99.2  | 69.5      | 87.2   | 94.5      | 61           |
> | intern2-7b-RM<br/>+ outcome labels | ORM         | 89.6        | 87.71 | 84.21     | 92.43  | 94.06     | 63.4         |
> | OpenPRM<br/>(intern)               | Last Step   | 91.1        | 98.0  | 81.6      | 89.5   | 95.1      | 68.1         |
> |                                    | Min Step    | 91.9        | 96.7  | 83.6      | 91.6   | 95.7      | 68.1         |
> |                                    | Max Step    | 88.7        | 95.0  | 80.5      | 88.2   | 91.1      | 68.1         |
> |                                    | Simple Avg  | 91.3        | 97.2  | 83.3      | 89.9   | 94.8      | 68.1         |
> |                                    | Weight Avg  | 91.4        | 98.0  | 82.7      | 89.7   | 95.0      | 68.1         |
> |                                    | Dynamic     | 91.6        | 97.8  | 83.3      | 90.4   | 95.0      | 68.1         |
> |                                    | Min Delta   | 77.8        | 64.3  | 75.7      | 76.4   | 95.1      | 68.1         |
> |                                    | Max Delta   | 77.7        | 77.4  | 69.5      | 74.3   | 89.5      | 68.1         |
>
> [1] Reward Model Ensembles Help Mitigate Overoptimization. ICLR 2024
>
> [2] Rewarding Progress: Scaling Automated Process Verifiers for LLM Reasoning. Arxiv 2024

---

> ### Author Response · Authors · 2024-11-23
> **Response - 4**
>
> ### Q2: Review of Previous Methods (Including MSE Loss)
>
> We appreciate your comments regarding the review of MSE loss and its application in reward models (RMs) and Step-DPO. Here are our thoughts:
>
> **A1. MSE Loss for PRMs:**
> While most Outcome-level Reward Models (RMs) are traditionally trained using the Bradley-Terry objective, Process Reward Models (PRMs) often utilize MSE loss for optimization. This approach is particularly suited to PRMs as they assign probabilities to individual steps in a sequence for questions with specific labels like math and code. For instance, given labels such as $(1, 1, 0, 0)$ for a sample with steps $(s_1, s_2, s_3, s_4)$, the PRM’s goal is to predict the probability of correctness for each step. The corresponding training objective can be expressed as:
> $$\mathcal{L} = \sum_{t}^N \left(r(s_{0:t} \mid x) - y_t\right)^2$$
>
> Here:
> - $r(s_{0:t} \mid x)$ represents the predicted reward probability for the sequence of steps up to $t$, given the input $x$.
> - $y_t$ is the ground-truth label for step $t$.
>
> This formulation ensures that the PRM learns to align its step-wise predictions closely with the provided labels, thereby improving process-level performance.
>
> -----
>
> Recent works such as OVM [1] and AutoPSV [2], have explored MSE loss-based ORMs. Their training objective is:
> $$\mathcal{L}{\text{total}}(Q) = \frac{1}{|Q|} \sum{q \in Q} \frac{1}{n} \sum_{i=1}^n \sum_{t=1}^{m_i} (f_{\theta} (S_i^{(1:t)};q) - y_i)^2$$
>
> where $n$ represents the number of solutions per question, and $m_i$ is the number of steps in the $i$-th solution. The output approximates the expected correctness probability for any partial content:
> $$f_{\theta}(S^{(1:t)};q)$$
> Details of this derivation are provided in OVM [1]. Additionally, recent studies [3] have shown that binary classification models can be equivalent to BT models, further supporting the viability of MSE-based regression models.
>
> **A2. Step-DPO.**
> We acknowledge the controversy regarding Step-DPO. While it does not explicitly define reward models, DPO models can be considered a type of implicit reward model [4]. They can assign outcome- or process-level rewards to samples, formulated as:
>  $$r(x,y) = \beta \log \cfrac{\pi_{\theta} (y|x)}{\pi_{\text{ref}} (y|x)} + \beta \log Z(x)$$
>
> [1] OVM, Outcome-supervised Value Models for Planning in Mathematical Reasoning. NAACL
>
> [2] AutoPSV: Automated Process-Supervised Verifier. NeurIPS 2024
>
> [3] Rethinking Bradley-Terry Models in Preference-Based Reward Modeling: Foundations, Theory, and Alternatives. Arxiv 2024
>
> [4] Direct Preference Optimization: Your Language Model is Secretly a Reward Model. NeurIPS 2023
>
> ### Q3: Computational Efficiency of the Proposed Method
> We compared the computational efficiency of our method with MCTS-based approaches under similar sampling budgets. The experimental setup and results are as follows:
>
> **A1. Sampling Settings:**
> -  **OpenPRM:** We start from 60k samples, with 64 responses per sample, producing approximately 3.84M question-answer pairs in 24 hours. From this, 90k pairs were selected for training.
> - **MCTS:** We start from 10k samples, with 4 responses per sample. Responses were split into sentences, resulting in 240k partial outputs. Sampling 8 full paths for each pair of partial outputs produced ~3.84M question-answer pairs in 24 hours. Of these, 97k pairs were used for training.
>
> **A2. Results:**
> The results of PRM trained with OpenPRM and MCTS are shown below:
>
> | Models                             | Aggregation | RewardBench |       |           |        |           | PRM800k Test |
> | ---------------------------------- | ----------- | ----------- | ----- | --------- | ------ | --------- | ------------ |
> |                                    |             | Overall     | Chat  | Chat Hard | Safety | Reasoning |              |
> | intern2-7b-RM                      | ORM         | 87.6        | 99.2  | 69.5      | 87.2   | 94.5      | 61           |
> | intern2-7b-RM<br/>+ outcome labels | ORM         | 89.6        | 87.71 | 84.21     | 92.43  | 94.06     | 63.4         |
> | OpenPRM (intern)                   | Min Step    | 91.9        | 96.7  | 83.6      | 91.6   | 95.7      | 68.1         |
> | MCTS (intern)                      | Min Step    | 91.4        | 95.5  | 81.6      | 93.2   | 95.4      | 68.2         |
>
> In fact, the MCTS method remains a powerful baseline but is significantly more resource-intensive, requiring up to 10 times the computational cost of our approach. With a larger sampling budget, the MCTS method could still be effectively utilized. However, our method offers a more efficient alternative and can be further optimized with more accurate similarity computation techniques, such as embedding-based methods. While these approaches may increase the time required to construct the tree, they hold great potential for improving performance.
>
> We plan to explore these optimizations in future work to further enhance the efficiency and accuracy of our method.

---

### Official Review · Reviewer_WSMi · 2024-11-11

**Soundness:** 4
**Presentation:** 3
**Contribution:** 4
**Rating:** 8
**Confidence:** 4

**Summary:**

This paper presents OpenPRM, a reward model designed to incorporate both outcome-based and process-based rewards. The authors leverage outcome-based reward models to generate sentence-level preference trees, subsequently backpropagating these outcome rewards to provide weak supervision for process rewards. By jointly training the model on both types of signals, OpenPRM achieves huge improvements over existing ORMs and PRMs. Additionally, the authors demonstrate that OpenPRM can be effectively used during inference to guide the decoding process, with model performance scaling positively with the number of samples.

**Strengths:**

This model could provide substantial value to the RLHF research community. Although similar ideas have been extensively explored in prior work, particularly in domains like math and coding that focus on reasoning, few of those works released their models and datasets. By extending to more open-domain tasks, this model could facilitate RLHF research beyond specialized tasks.

**Weaknesses:**

One minor weakness is the lack of novel technical contributions; however, given the value this model brings to the RLHF research community, this is not a major concern. A more significant weakness is the lack of detailed analysis of the experimental results. While the authors demonstrate that OpenPRM generally outperforms existing reward models, they do not provide a breakdown of the results. Specifically:
1. The improved outcome-level reward performance is not clearly attributed—it's uncertain whether the gains come from the method itself or simply from better data. The data used (including UF, HS2, and additional math/coding datasets) may be better than that used in FsFairx or internRM. It would be helpful to see a baseline where these models are fine-tuned on the same data but with only outcome-level supervision to isolate the impact of the method.
2. Although OpenPRM shows better results overall, its performance on Chat is consistently lower. An analysis explaining this discrepancy would strengthen the results.
3. Figure 3 highlights the scaling effect of BoN, but no analysis is provided on PBM, which would provide a more complete picture.

Additional minor issues include:
1. Figure 3 lacks clarity due to overlapping lines, making it difficult to interpret the results.
2. The authors might consider evaluating with the OffsetBias dataset, which could offer a more fine-grained assessment of bias in reward models.

**Questions:**

See weaknesses.

---

> ### Author Response · Authors · 2024-11-23
> **Response - 1**
>
> We sincerely thank you for your thoughtful feedback and the time and effort you have dedicated to reviewing our work. Below, we address your queries and provide additional clarifications where necessary.
>
> ### Q1: Ablation Study on Outcome-Level Reward Model Performance
>
> We consider the backbone models (Intern-RM, FsFairX-RM) as baselines, further fine-tuning these models using datasets with both outcome-level and process-level labels. For the ablation study, the dataset is divided into preference tree pairs (process-level) and outcome pairs (outcome-level).
>
> As illustrated in Figure 5(a), our results demonstrate that OpenPRM, trained on our dataset with process-level labels (Preference Tree Pairs), outperforms models directly fine-tuned on data with outcome-level labels. This highlights the efficacy of process-level training in improving performance.
>
> ### Q2: Performance of OpenPRM in the Chat Category
>
> Thank you for highlighting this point. As shown in Table 2, OpenPRM’s scores in the Chat category decrease, while performance in the Chat Hard category improves. This reflects the inherent trade-off between optimizing for simpler conversational tasks (Chat Easy) and more complex reasoning tasks (Chat Hard), which is a known challenge in reward model optimization, as also noted in RewardBench [1].
>
> The distinction between these categories lies in their data sources: the Chat category includes datasets like AlpacaEval and MT Bench, while Chat Hard is derived from MT Bench and LLMBar. The decreased scores in the Chat category are largely due to AlpacaEval, as there is a distributional shift between this dataset and our training data, which is sourced from UltraFeedback.
>
> [1] Lambert, Nathan, et al. “RewardBench: Evaluating reward models for language modeling.” _arXiv preprint arXiv:2403.13787_ (2024).
>
> ### Q3: Scaling Effects on Process Beam Search
>
> We have analyzed the scaling effects of Process Beam Search (PBS), where beam search is conducted at the sentence level, selecting the top N generated outputs based on PRM rewards.
> As presented in the following table, the results indicate that OpenPRM consistently outperforms Bag of N-grams (BoN) in PBS settings, showcasing its effectiveness and reliability in these scenarios.
>
> | Models     | Setting       | GPQA  |       |       |       |       |       |       |       | MMLU-Pro |       |       |       |       |       |       |       |
> | ---------- | ------------- | ----- | ----- | ----- | ----- | ----- | ----- | ----- | ----- | -------- | ----- | ----- | ----- | ----- | ----- | ----- | ----- |
> |            |               | 1     | 2     | 4     | 8     | 16    | 32    | 64    | 128   | 1        | 2     | 4     | 8     | 16    | 32    | 64    | 128   |
> | -          | Coverage      | 24.34 | 39.39 | 57.58 | 72.32 | 84.95 | 92.12 | 96.87 | 98.99 | 37.20    | 51.24 | 62.12 | 72.00 | 79.80 | 86.56 | 91.44 | 94.6  |
> | -          | Majority Vote | 24.34 | 25.25 | 23.03 | 26.16 | 26.77 | 25.56 | 24.95 | 26.06 | 37.20    | 38.40 | 41.28 | 42.00 | 43.44 | 44.12 | 44.64 | 45    |
> | FsFairX-RM | ORM - BoN     | 24.34 | 28.79 | 29.39 | 28.69 | 26.77 | 24.04 | 23.03 | 22.12 | 37.20    | 41.00 | 42.16 | 42.44 | 43.52 | 43.24 | 42.16 | 42.2  |
> | Intern-RM  | ORM - BoN     | 24.34 | 30.1  | 31.72 | 32.32 | 31.92 | 30.81 | 31.52 | 27.98 | 37.2     | 42.64 | 45.36 | 46.56 | 48.12 | 48.52 | 48.88 | 48.48 |
> | OpenPRM    | PRM - BoN     | 24.34 | 28.59 | 30    | 31.92 | 32.63 | 33.13 | 34.34 | 37.17 | 37.20    | 42.84 | 46.04 | 46.88 | 49.56 | 50.52 | 50.32 | 50.28 |
> | OpenPRM    | PRM - PBS     | 24.34 | 31.82 | 32.83 | 36.87 | -     | -     | -     | -     | 37.20    | 43.8  | 46.2  | 51.4  | -     | -     | -     | -     |
>
> However, we also observed significant variance in the scaling effect of PBS across different tasks, in contrast to the more consistent scaling effect seen with best-of-N methods. This variance can likely be attributed to differences in the data distributions across tasks, which highlight the need for further investigation into handling data mixtures effectively.
> We plan to continue exploring this issue in future work to better understand and address these challenges.

---

> ### Author Response · Authors · 2024-11-23
> **Response - 2**
>
> ### Q4: Additional Results on OffsetBias Datasets
>
> In addition to benchmarks like RewardBench, UltraFeedback, and PRM800k, we evaluated our reward models using OffsetBias [1], which provides a more granular assessment of bias in reward models.
>
> The results are summarized in the table below, showcasing the effectiveness of OpenPRM across various bias metrics:
>
> | Models                             | OffsetBias   |                          |                     |                                        |             |                        |         |
> | ---------------------------------- | ------------ | ------------------------ | ------------------- | -------------------------------------- | ----------- | ---------------------- | ------- |
> |                                    | Concreteness | Content<br/>Continuation | Empty<br/>Reference | Faimilar Knowledge<br/>Preference Bias | Length Bias | Nested<br/>Instruction | Overall |
> | Eurus-RM-7B                        | 71.4         | 66.7                     | 84.6                | 33.3                                   | 41.2        | 66.7                   | 60      |
> | FsfairX-LLaMA3-RM                  | 100          | 91.7                     | 53.8                | 91.7                                   | 41.2        | 58.3                   | 71.3    |
> | FsfairX-LLaMA3-RM<br/>+OffsetBias  | 92.9         | 100                      | 46.2                | 58.3                                   | 82.4        | 83.3                   | 77.5    |
> | LLaMA3-8B-Instruct<br/>+OffsetBias | 100          | 95.8                     | 92.3                | 83.3                                   | 85.3        | 50                     | 85      |
> | Intern2-7b-RM                      | 1            | 1                        | 1                   | 58.3                                   | 58.8        | 91.7                   | 84.8    |
> | Intern2-7b-RM<br/>+ OpenPRM (Our)  | 92.86        | 83.3                     | 1                   | 83.3                                   | 88.2        | 91.7                   | 89.9    |
>
> [1] Park, Junsoo, et al. "Offsetbias: Leveraging debiased data for tuning evaluators." arXiv preprint arXiv:2407.06551 (2024).
>
> ### Q5: Clarity of Figure 3
>
> Thank you for this suggestion. We have revised Figure 3 to better visualize the BoN effects across different reward models. The updated graph now separately highlights the performance of our RM and includes detailed coverage curves for enhanced clarity.
>
> ----
>
> Once again, we appreciate your valuable comments and constructive suggestions, which have helped us improve the quality and clarity of our work.

---

> > ### Comment · Reviewer_WSMi · 2024-11-23
> >
> > The authors have adequately addressed my concerns. I raise my score to accept.

---

### Author Response · Authors · 2024-11-28
**Revision Summary**

Dear Reviewers,

Thank you for your thoughtful comments and constructive feedback on our manuscript. We have carefully revised the paper to address your concerns, incorporating additional explanations, analyses, and clarifications as needed. To facilitate your review, all new content in the revised manuscript is highlighted in blue. Below, we provide a summary of the key changes made in response to your comments:

- Discussion of results in the Chat Category (@Reviewer WSMi) : We have expanded the discussion regarding the performance changes in the Chat Category, which can now be found in Appendix D.1.
 - Scaling effects on process beam search (@Reviewer WSMi and  MtX7): Additional results on the scaling effects of process beam search are included in Appendix D.4 (Figure 7).
- Explanation of ORM results with different aggregation strategies (@Reviewer SVaF and 4T3D): We have added explanations and further results for various aggregation strategies, presented in Appendix D.1 (Table 6).
- Results on Offsetbias datasets (@Reviewer WSMi): We have included new results for the Offsetbias dataset, provided in Appendix D.2 (Table 7).
- Details on dataset construction (@Reviewer SVaF and 4T3D): Additional details regarding the selection of edit distance are illustrated in Figure 5, and a real-world example of tree building is included in Appendix B.3 (Table 5).
- Comparison with MCTS methods (@Reviewer SVaF and MtX7): We have conducted an analysis and comparison with MCTS methods, which are now detailed in Appendix D.3 (Table 8).
- Training data details (@Reviewer 4T3D): Information about the training data format has been added in Appendix B.1.
- Updated visualization of scaling effects on BoN (@Reviewer WSMi): To enhance clarity, the results in Figure 3 have been divided into coverage curves and scaling effects of RMs, and we have updated the visualization style in Figures 3, 8, 9, 10, 11, and 12.

We hope these revisions address your concerns and improve the clarity and comprehensiveness of our manuscript. Thank you once again for your time and consideration.

Sincerely,

Authors

---

### Meta-Review · Area_Chair_XMv3 · 2024-12-22

**Metareview:**

This paper proposes a method for building process-based reward models for instruction following models. The authors use outcome-based reward models to build weak sentence-level preference trees based on prefix similarity. Experiments show that the proposed method results in better reward models and can be used at inference time to improve decoding.

Strengths:
1. This work extends process-based reward models from the math domain to general domains and can be potentially applied to many applications.

Weaknesses:
1. MCTS gets a similar performance as this work, although this work is more computationally efficient.

Overall, the proposed approach has a wide range of potential applications in language model alignment and I don't see significant weaknesses of this paper. Based on reviewer ratings, I recommend acceptance. However, I wouldn't mind if the paper gets rejected.

**Additional Comments On Reviewer Discussion:**

One reviewer pointed out the comparison to MCTS, and authors conducted additional experiments, which shows MCTS as a very strong baseline. I'd recommend incorporating this result into the next version of the paper.

---

### Decision · Program_Chairs · 2025-01-22

Accept (Poster)